# Unravelling druggable signalling networks that control F508del-CFTR proteostasis

**Ramanath Narayana Hegde**[1,2†], **Seetharaman Parashuraman**[1,2*†], **Francesco Iorio**[2‡§], **Fabiana Ciciriello**[2,3,4‡], **Fabrizio Capuani**[2¶], **Annamaria Carissimo**[2], **Diego Carrella**[2], **Vincenzo Belcastro**[2], **Advait Subramanian**[1], **Laura Bounti**[1**], **Maria Persico**[2], **Graeme Carlile**[4], **Luis Galietta**[5], **David Y Thomas**[4], **Diego Di Bernardo**[2,6], **Alberto Luini**[1,2,7*]

[1]Institute of Protein Biochemistry, National Research Council, Naples, Italy; [2]Telethon Institute of Genetics and Medicine, Pozzuoli, Italy; [3]Biology and Biotechnology Department "Charles Darwin", Sapienza University, Rome, Italy; [4]Department of Biochemistry, McIntyre Medical Sciences Building, McGill University, Montréal, Canada; [5]U.O.C. Genetica Medica, Institute of Giannina Gaslini, Genova, Italy; [6]Department of Electrical Engineering and Information Technology, University of Naples Federico II, Naples, Italy; [7]Istituto di Ricovero e Cura a Carattere Scientifico SDN, Naples, Italy

**\*For correspondence:**
r.parashuraman@ibp.cnr.it (SP);
a.luini@ibp.cnr.it (AL)

[†]These authors contributed equally to this work
[‡]These authors also contributed equally to this work

**Present address:** [§]European Molecular Biology Laboratory – European Bioinformatics Institute, Wellcome Trust Genome Campus, Cambridge, United Kingdom; [¶]Department of Physics, University of Rome "La Sapienza", Rome, Italy; [**]KU Leuven University, Leuven, Belgium

**Competing interests:** The authors declare that no competing interests exist.

**Abstract** Cystic fibrosis (CF) is caused by mutations in CF transmembrane conductance regulator (CFTR). The most frequent mutation (F508del-CFTR) results in altered proteostasis, that is, in the misfolding and intracellular degradation of the protein. The F508del-CFTR proteostasis machinery and its homeostatic regulation are well studied, while the question whether 'classical' signalling pathways and phosphorylation cascades might control proteostasis remains barely explored. Here, we have unravelled signalling cascades acting selectively on the F508del-CFTR folding-trafficking defects by analysing the mechanisms of action of F508del-CFTR proteostasis regulator drugs through an approach based on transcriptional profiling followed by deconvolution of their gene signatures. Targeting multiple components of these signalling pathways resulted in potent and specific correction of F508del-CFTR proteostasis and in synergy with pharmacochaperones. These results provide new insights into the physiology of cellular proteostasis and a rational basis for developing effective pharmacological correctors of the F508del-CFTR defect.

## Introduction

Cystic fibrosis (CF) is the most common lethal genetic disease in Caucasians. It is caused by mutations in the *CF transmembrane conductance regulator (CFTR)* gene that encodes a chloride channel localised to the apical membrane of several epithelial cells. Mutations that cause CFTR loss of function impair the transepithelial movement of salts at the cell surface, resulting in pleiotropic organ pathology and, in the lungs, in chronic bacterial infections that eventually lead to organ fibrosis and failure (*Riordan, 2008*).

The CFTR protein comprises two membrane-spanning domains, two cytosolic nucleotide-binding domains, and a regulatory domain, folded together into a channel (*Riordan, 2008*). Folding occurs in the endoplasmic reticulum (ER) through the sequential action of multiple chaperone complexes (*Loo et al., 1998*; *Meacham et al., 1999*; *Rosser et al., 2008*) and is followed by export out of the ER and glycosylation in the Golgi before arrival at the plasma membrane (PM), where CFTR

**eLife digest** Cystic fibrosis is a genetic disease that commonly affects people of European descent. The condition is caused by mutations in the gene encoding a protein called "cystic fibrosis transmembrane conductance regulator" (or CFTR for short). CFTR forms a channel in the membrane of cells in the lungs that help transport salt across the membrane. Mutated versions of the protein are not as efficient at transporting salts, and eventually this damages the lung tissue. As the damage progresses, individuals become very vulnerable to bacterial infections that further damage the lungs and may eventually lead to death.

One of the reasons CFTR mutations are harmful is that they cause the protein to fold up incorrectly and remain trapped inside the cell. Cells have quality control systems that recognize and destroy poorly folded proteins, and so only a few of the mutated CFTR proteins ever make it to the membrane to move salts. New therapies have been developed that improve folding of the protein and/or help the CFTR proteins that make it to the membrane work better. But more and better treatment options are needed.

Hegde, Parashuraman et al. have now tested drugs that control how proteins fold and move to the membrane to see how they affect gene expression in cells with the most common cystic fibrosis-causing mutation. These drugs are known to improve the activity of the CFTR mutant, but do so too weakly to be of clinical interest. The experiments revealed that the expression of a few hundred genes was changed in response the drugs. Many of these genes were involved in major signalling pathways that control how CFTR is folded and trafficked within cells.

Next, Hegde, Parashuraman et al. tested drugs that inhibit these signalling pathways to see if they improve salt handling in the mutated cells. The experiments demonstrated that these inhibitor drugs efficiently block the breakdown of misfolded CFTR, or boost the likelihood of CFTR making it to the membrane, helping improve salt trafficking in the cells. The inhibitors produced even better results when used in combination with a known CFTR-protecting drug. The results suggest that identifying and targeting signalling pathways involved in the folding, trafficking, and breakdown of CFTR may prove a promising way to treat cystic fibrosis.

undergoes several cycles of endocytosis before degradation in the lysosomes (*Gentzsch et al., 2004*). The most frequent mutant, which is present in ~90% of the patients with CF, misses a phenyl-alanine at position 508 (F508del-CFTR) and folds in a kinetically and thermodynamically impaired fashion into a conformation that is recognized as defective by the ER quality control (ERQC) system. It is thus retained in the ER and targeted for ER-associated degradation (ERAD) by the ubiquitin–proteasome machinery (*Jensen et al., 1995*; *Ward et al., 1995*). A small fraction of F508del-CFTR may escape degradation in the ER and reach the PM, where it can function as a channel. This might have therapeutic relevance because patients that express even low levels of functional channel have milder symptoms (*Amaral, 2005*). However, at the PM, F508del-CFTR is recognized by the peripheral (or PM-associated) quality control (PQC) system and is rapidly degraded in the lysosomes (*Okiyoneda et al., 2010*).

In contrast to the fairly extensive knowledge about the machinery involved in the proteostasis (or protein homeostasis) of F508del-CFTR, the regulatory mechanisms that operate on the F508del-CFTR proteostasis machinery remain relatively less explored. Notable exceptions are the recent studies on the effects of the unfolded protein response and heat shock response (UPR and HSR, respectively) on the proteostasis of F508del-CFTR. The UPR and HSR operate as *homeostatic* reactions that tend to redress the imbalances between the load of unfolded proteins and the folding capacity of a cell essentially by enhancing the transcription of the cellular folding machinery. Investigators have therefore sought to induce these reactions by pharmacological means with the aim to rescue the F508del-CFTR folding/transport defect, with partial success (*Roth et al., 2014*; *Ryno et al., 2013*).

Very little is known instead about the regulation of proteostasis by the 'classical' signalling networks composed of GTPases, second messengers, kinases, etc. that are usually activated by PM receptors and control most, if not all, of the cellular functions. We and others have previously shown

that constitutive trafficking along the secretory pathway is potently controlled by such signalling networks triggered by both extra- and intracellular stimuli (*Cancino et al., 2014*; *Chia et al., 2012*; *De Matteis et al., 1993*; *Farhan et al., 2010*; *Giannotta et al., 2012*; *Pulvirenti et al., 2008*; *Simpson et al., 2012*). This suggests that the machinery of proteostasis viz. protein synthesis, folding, and degradation, is also likely to be controlled by similar signalling systems. Identifying the relevant regulatory components of these systems would not only enhance our understanding of the physiology of proteostasis, but also have significant impact on future therapeutic developments, because components of the signalling cascades, such as membrane receptors and kinases, are generally druggable, and are, in fact, the main targets of most known drugs.

Thus, this study aims to uncover signalling pathways that control proteostasis of F508del-CFTR. To this end, we have developed a strategy based on the analysis of the mechanisms of action (MOAs) of drugs that regulate the proteostasis of F508del-CFTR. The choice of this strategy over more traditional approaches such as kinome-wide screenings was based on the rationale that since many of the successful drugs target multiple molecular pathways simultaneously (*Lu et al., 2012*) and with limited toxicity, elucidating the MOAs of these drugs might lead to uncovering molecular networks that regulate proteostasis in a synergistic and relatively 'safe' manner.

Several drugs that regulate the proteostasis of F508del-CFTR (hereinafter referred to as proteostasis regulators) and enhance its ability to reach the PM have been identified over the years, largely through screening campaigns (*Calamini et al., 2012*; *Carlile et al., 2012*; *Hutt et al., 2010*). In addition, molecules that bind directly to F508del-CFTR and facilitate its folding have also been characterized (pharmacochaperones) (*Calamini et al., 2012*; *Kalid et al., 2010*; *Odolczyk et al., 2013*; *Pedemonte et al., 2005*; *Sampson et al., 2011*; *Van Goor et al., 2006*; *Wang et al., 2007*). Both these groups of drugs that enhance the ability of F508del-CFTR to reach the PM are referred to as correctors. The MOA of the pharmacochaperones has been partially understood (*Farinha et al., 2013a*; *Okiyoneda et al., 2013*), and they are approaching the level of effectiveness required for clinical use ([*Wainwright et al., 2015*] and see also http://www.fda.gov/NewsEvents/Newsroom/PressAnnouncements/ucm453565.htm), while the proteostasis regulators are presently too ineffective to be of clinical interest.

Here, we have analysed the MOAs corrector drugs that are proteostasis regulators by deconvolving their transcriptional effects. Changes in gene expression are significant components of the MOAs of many drugs (*Popescu, 2003*; *Santagata et al., 2013*), and the analysis of transcriptional MOAs is a growing research area (*Iorio et al., 2010*; *Iskar et al., 2013*). However, a major difficulty here is that the available proteostasis regulator drugs include representatives of diverse pharmacological classes such as histone deacetylase inhibitors (*Hutt et al., 2010*), poly(ADP-ribose) polymerase inhibitors (*Anjos et al., 2012*; *Carlile et al., 2012*), hormone receptor activators (*Caohuy et al., 2009*), cardiac glycosides (*Zhang et al., 2012*), and others. Thus, the effects of the available F508del-CFTR correctors are most probably not mediated by their heterogeneous principal MOA, but by some unknown weak secondary MOAs ('side effects') that these drugs share. In order to extract the transcriptional changes that are correction-related from those that are due to the (correction-irrelevant) principal MOAs of the corrector drugs we have developed a new approach based on the 'fuzzy' intersection of gene expression profiles. This method, applied to a set of proteostatic correctors will identify genes that are commonly modified by these drugs and should therefore correspond to the correction-related pathways and not to their heterogeneous primary effects. Using this strategy, we harvested a group of few hundred genes that are regulated by most of the proteostatic correctors, and then derived a series of molecular networks from this gene pool through bioinformatic and experimental approaches. Several of these networks are signalling pathways made up of druggable receptors and kinases. Silencing or targeting these pathways with chemical blockers inhibit the degradation in the ER and enhance the transport of F508del-CFTR to the PM. Moreover, the large pool of ER-localised foldable F508del-CFTR that results from the inhibition of ER degradation can be acted upon by pharmacochaperones, to further enhance correction. These findings build on previous screening studies and on the accumulated knowledge of F508del-CFTR proteostasis to start to define the network of signalling pathways that control F508del-CFTR proteostasis, and thereby provide a rational basis for the development of novel, potent and specific proteostasis corrector treatments for CF.

## Results

### Proteostasis correctors have a shared transcriptional signature

Proteostasis regulators share the ability to correct (albeit weakly) the F508del-CFTR folding-trafficking defect but have principal pharmacological effects not related to F508del-CFTR correction. If the correction-related MOAs of these drugs are transcription-dependent (see Materials and methods for evidence that they are), then the gene signatures of these drugs should comprise both genes related to F508del-CFTR correction and genes related to their heterogeneous primary effects.

Thus, we sought to analyse the transcriptional MOAs of correctors (24 drugs/conditions altogether) with different chemical structures and pharmacological activities (*Table 1*), excluding known pharmacochaperones. The gene signatures of 13 correctors were obtained in our laboratories using immortalised CF bronchial epithelial (CFBE41o-) cells (*Kunzelmann et al., 1993*) on an Agilent microarray platform (CFBE dataset; see *Supplementary file 1* and GEO accession number GSE67698 for the expression profiles). Another 11 signatures were extracted from the MANTRA (Mode of action by network analysis, [*Iorio et al., 2010*]; MANTRA dataset) that were based on the Affymetrix platform. Two drugs, glafenine and ouabain, were present in both datasets. Even though the signatures of each drug in the two platforms were obtained from different cell lines, they were similar enough (not shown) to suggest that two datasets can be treated together.

To extract the correction-related transcriptional effects from those due to primary effects of the correctors, we first attempted to cluster the drugs based on commonalities in their transcriptional profiles. These attempts using classical and alternate clustering methods did not yield meaningful results (see *Figure 1—figure supplement 1* and *Supplementary file 2*), possibly because the strong transcriptional effects of the heterogeneous principal MOAs of these drugs obscures the potential clustering of drug signatures based on their secondary correction-relevant MOAs.

In order to detect these weak but common transcriptional signatures we developed a method based on the fuzzy intersection of transcriptional profiles (FIT) (*Figure 1A*). Here, the corrector gene signatures are 'intersected' to identify their commonalities, and this returned a pool of genes that are potentially correction-related (CORE) and are modulated by most of the correctors. The intersections among the majority of the signatures should include the correction-related (CORE) genes but exclude genes related to the heterogeneous principal effects of the drugs. The method thus captures common MOAs but not MOAs specific for individual drugs or small groups of drugs.

The main parameters of the FIT analysis (number of correctors; number of genes to be analysed in each signature, and cut-off threshold for inclusion in the correction-relevant gene pool; see Materials and methods and *Figure 1B–C*) were selected to identify a sufficiently large CORE gene pool for pathway analysis, and also to minimise the number of 'false' CORE genes. The FIT analysis of the gene signatures resulted in 219 downregulated and 402 upregulated CORE genes (*Supplementary file 3*; *Figure 1D*; see also Materials and methods). Each of these CORE genes were shared by 70% of the corrector signatures. The number of CORE genes were threefold higher than that expected on a random basis (see Materials and methods).

This indicates that common transcriptional programmes that might be correction-relevant are indeed embedded in the signatures of proteostasis correctors.

### Identification of CORE genes/pathways involved in F508del-CFTR correction

To understand the relation of CORE genes to CFTR proteostasis, we built a dataset of known F508del-CFTR proteostasis-relevant genes by assembling literature data (*Supplementary file 4*) and mapped their interactions with the CORE pool using STRING (*Franceschini et al., 2013*). We found extensive and statistically significant (see Materials and methods) protein-protein interactions among the nodes of the union of these two datasets (*Figure 1F*), indicating that (at least a fraction of) the CORE genes are related to CFTR proteostasis. Significant interactions were also found between the CORE genes from CFBE and the MANTRA datasets (not shown) confirming they are related and thus can be analysed together.

We next applied standard bioinformatic tools to the CORE gene pool to identify functionally coherent pathways/networks/groups. A search proteostasis components among CORE genes retrieved 48 folding/degradation and 24 transport-machinery components (*Supplementary file 5*),

**Table 1.** The list of corrector drugs used in this study with their corresponding known primary MOAs (related to *Figure 1*).

| Drugs of the CFBE dataset (Reference for correction activity) | Primary Use/Class |
|---|---|
| 4-AN, PARP1 inhibitor (*Anjos et al., 2012*) | PARP1 inhibitor |
| ABT888 (*Anjos et al., 2012*) | A poly(ADP-ribose) polymerase (PARP) -1 and -2 inhibitor with chemosensitizing and antitumor activities. ABT-888 inhibits PARPs, thereby inhibiting DNA repair and potentiating the cytotoxicity of DNA-damaging agents. |
| Glafenine (*Robert et al., 2010*) | An anthranilic acid derivative with analgesic properties used for the relief of all types of pain (1) |
| GSK339 (DY Thomas lab, unpublished) | Androgen receptor ligand (*Norris et al., 2009*). |
| Ibuprofen (*Carlile et al., 2015*) | Ibuprofen is a nonsteroidal anti-inflammatory drug. It is a non-selective inhibitor of cyclooxygenase. |
| JFD03094 | PARP inhibitor |
| KM11060 (*Robert et al., 2008*) | PDE5 inhibitor (an analog of sildenafil). |
| Latonduine (*Carlile et al., 2012*) | PARP3 inhibitor |
| Minocycline H (D Y Thomas lab unpublished) | A tetracycline analog that inhibits protein synthesis in bacteria. Also known to inhibit 5-lipooxygenase in the brain (2). |
| Ouabagenin (*Zhang et al., 2012*) | A cardioactive glycoside obtained from the seeds of *Strophanthus gratus*. Acts by inhibiting $Na^+/K^+$-ATPase, resulting in an increase in intracellular sodium and calcium concentrations (2). |
| Ouabain (*Zhang et al., 2012*) | A cardioactive glycoside obtained from the seeds of *Strophanthus gratus*. Acts by inhibiting $Na^+/K^+$-ATPase, resulting in an increase in intracellular sodium and calcium concentrations (2). |
| PJ34 (*Anjos et al., 2012*) | PARP1 inhibitor |
| Low temperature (*Denning et al., 1992*) | |
| Chloramphenicol (*Carlile et al., 2007*) | Inhibitor bacterial protein synthesis by binding to 23S rRNA and preventing peptidyl transferase activity (2). |
| Chlorzoxazone (*Carlile et al., 2007*) | Muscle relaxant. Acts by inhibiting degranulation of mast cells and preventing the release of histamine and slow-reacting substance of anaphylaxis. It acts at the level of the spinal cord and subcortical areas of the brain where it inhibits multi-synaptic reflex arcs involved in producing and maintaining skeletal muscle spasm (2). |
| Dexamethasone (*Caohuy et al., 2009*) | A synthetic glucocorticoid agonist. Its anti-inflammatory properties are thought to involve phospholipase $A_2$ inhibitory proteins, lipocortins (2). |
| Doxorubicin (*Maitra et al., 2001*) | DNA intercalator that inhibits topoisomerase II activity by stabilizing the DNA-topoisomerase II complex (2). |
| Glafenine (*Robert et al., 2010*) | An anthranilic acid derivative with analgesic properties used for the relief of all types of pain (1). |
| Liothyronine (*Carlile et al., 2007*) | L-triiodothyronine (T3, liothyronine) thyroid hormone is normally synthesized and secreted by the thyroid gland. Most T3 is derived from peripheral monodeiodination of T4 (L-tetraiodothyronine, levothyroxine, L-thyroxine). The hormone finally delivered and used by the tissues is mainly T3. Liothyronine acts on the body to increase the basal metabolic rate, affect protein synthesis and increase the body's sensitivity to catecholamines (such as adrenaline). It is used to treat hypothyroidism (2). |
| MS-275 (*Hutt et al., 2010*) | Also known as Entinostat. An inhibitor of Class I histone deacetylases (preferentially HDAC 1, also HDAC 3) (*Hu et al., 2003*). |
| Scriptaid (*Hutt et al., 2010*) | An inhibitor of Class I histone deacetylases (HDAC1, HDAC3 and HDAC8) (*Hu et al., 2003*). |
| Strophanthidin (*Carlile et al., 2007*) | A cardioactive glycoside that inhibits $Na^+/K^+$-ATPase. Also known to inhibit the interaction of MDM2 and MDMX (1). |
| Thapsigargin (*Egan et al., 2002*) | A sesquiterpene lactone found in roots of *Thapsia garganica*. A non-competitive inhibitor of sarco/endoplasmic $Ca^{2+}$-ATPase (SERCA) (1). |
| Trichostatin-A (*Hutt et al., 2010*) | An inhibitor of histone deacetylases (HDAC1, HDAC3, HDAC8 and HDAC7) (*Hu et al., 2003*). |

(1) http://pubchem.ncbi.nlm.nih.gov/
(2) www.drugbank.ca

some of which are known to be involved in F508del-CFTR proteostasis. However, perhaps surprisingly, they were not significantly enriched, indicating that changes in the expression levels of significant numbers of the proteostasis machinery genes is not part of the MOA of the proteostasis regulator drugs. Potential explanations for this could be that some proteostasis genes that are

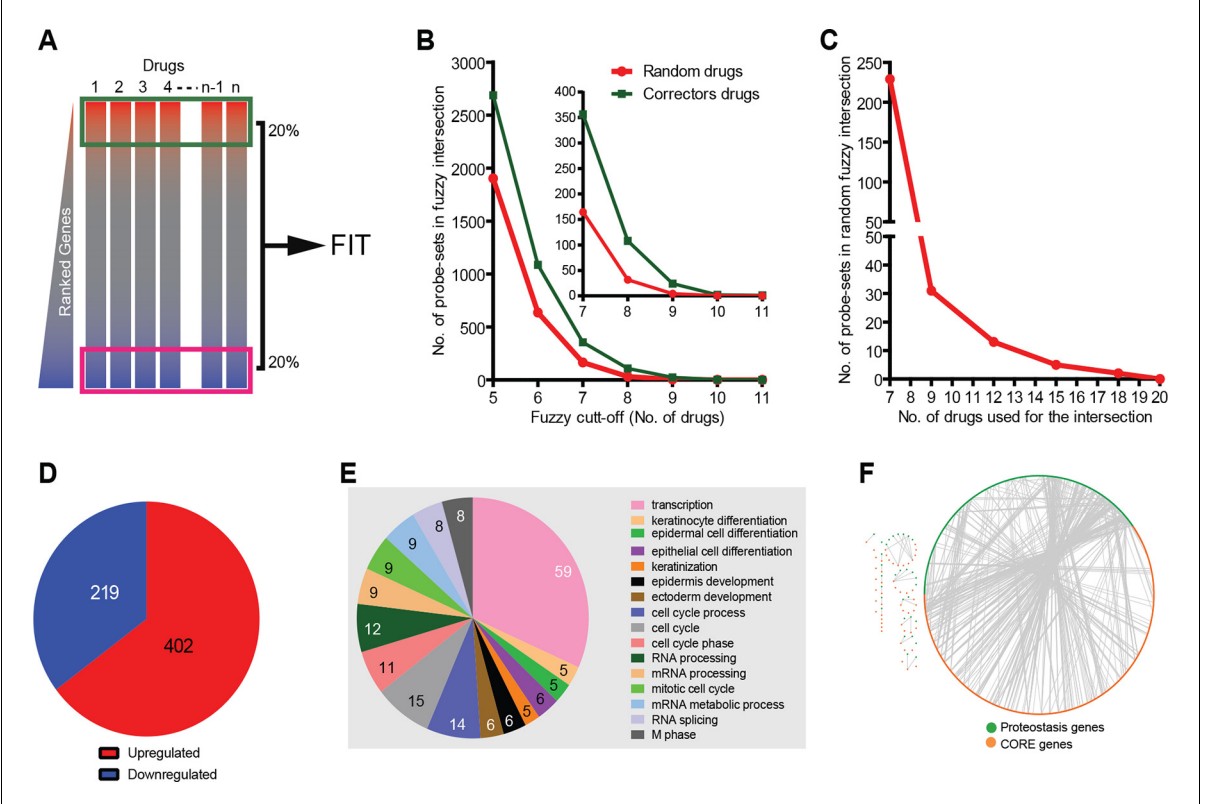

**Figure 1.** Corrector drugs modulate a set of CORE genes. (**A**) Schema of the FIT method. The upregulated (red) and downregulated genes (blue) were fuzzy intersected to identify CORE genes. (**B**) The number of probe sets in the corrector drug profiles (MANTRA dataset) as well as random profiles from MANTRA database were intersected with variable fuzzy cut-offs (represented as number of drugs out of 11) to obtain optimal fuzzy cut-off for the analysis. The enlargement (inset) shows that at the optimal fuzzy cut-off (0.7; 8 out of 11 drugs), the signal-to-noise ratio was close to 3 (108 probe-sets in the corrector drug intersection vs 32 in the random drug intersection). (**C**) At a fuzzy cut-off of 0.7, the number of random drug profiles used was varied, and the number of probe-sets present in the intersection is shown. (**D**) Using the optimal parameters (see **A, C** ) the FIT analysis resulted in 402 upregulated and 219 downregulated CORE genes. (**E**) The number of CORE genes associated with the enriched GO terms is shown. Those genes that did not associate with enriched GO terms were excluded from the chart. (**F**) Protein-protein interactions between the CORE and the proteostasis genes (restricted to those that connect the two groups) are shown.

The following figure supplements are available for figure 1:

**Figure supplement 1.** Corrector drugs (MANTRA dataset) have diverse transcriptional responses corresponding to their primary MOA.

**Figure supplement 2.** IPA based analysis uncovers networks of CORE genes.

crucial enough to have a strong effect on F508del-CFTR proteostasis are indeed regulated by the correctors, but are not numerous enough to result in a statistical enrichment of this gene group; or that the corrector drugs act by modulating the expression of regulatory genes/pathways that act post-translationally on the proteostasis machinery (see results from the screening below).

Since our interest was in the identification of signaling networks that regulate proteostasis we also searched for the presence of signalling molecules among CORE genes and found 24 kinases and 6 phosphatases (*Supplementary file 5*). Further, Ingenuity pathway analysis (IPA) tool identified several statistically significant signalling networks. The IPA networks comprised also (predicted) interactors of CORE genes, some of which were network hubs (*Figure 1—figure supplement 2*). Such hubs were often constituents of signalling pathways such as growth-factor-mediated pathways (e.g., receptors for vascular endothelial growth factor [VEGF] and platelet-derived growth factor [PDGF], phosphatidylinositol 3-kinase [PI3K], and mitogen-activated protein kinases [MAPKs]), inflammation-associated pathways (NF-κB subunits, Toll-like receptor 4 [TLR4]), stress-activated protein kinase (SAPK) pathways [MAP2K3/6 (MKK3/6), MAP2K4/7 (MKK4/7)], and casein-kinase pathway

(CSNK2A1/ CKII). These hubs might control the CORE genes. Of note, many of the hubs were frequently present in the gene signatures of the individual correctors, although below the fuzzy cut-off threshold of 0.7 required for inclusion in the CORE gene pool itself (not shown).

Analysis of the promoters of CORE genes aimed at the identification of upstream transcription factors did not generate interpretable results.

We then turned to experimental validation of the role of CORE genes in the regulation of F508del-CFTR proteostasis. Experiments were carried out using a characterised biochemical assay (See Materials and methods and *Figure 2—figure supplement 1* for details) that detects both the amount of core-glycosylated CFTR trapped in the ER (band B with western blotting) and the amount of CFTR fully glycosylated in the Golgi (most of which presumably resides at the PM; band C with western blotting). As a model system, we used non-polarised CFBE41o-cells stably expressing F508del-CFTR (*Bebok et al., 2005*) (hereafter referred to as CFBE); but many experiments were carried out also in HeLa, BHK and polarized CFBE cells, with results that were in good qualitative agreement with the CFBE data. While this assay is not suitable for large-scale screening, it provides quantitative information on the main proteostasis parameters including CFTR accumulation in the ER, ER-associated CFTR degradation, and transport and processing in the Golgi complex. Moreover, this assay is specific for proteostasis as it separates the effects on the F508del-CFTR protein from the effects on conductance as revealed by faster chloride-permeability assays (*Pedemonte et al., 2005*). Experimental validation was restricted to a limited set of genes: downregulated CORE genes (to exploit the availability of siRNA-based downregulation and of small-molecule inhibitors) that showed functional coherence, that is, were found in protein-protein interaction networks or in enriched GO groups; or were network hubs from Ingenuity analysis, or ubiquitin ligases and signalling molecules. In total, this resulted in a group of 108 genes (*Supplementary file 3*). Notably, these genes had no previously reported role in the regulation of F508del-CFTR proteostasis.

CFBE cells were treated with siRNAs against these genes and the effects on both bands B and C were monitored. As a reference for correction, we used the investigational drug VX-809 (*Van Goor et al., 2006*), a robust corrector that acts as a pharmacochaperone. VX-809 treatment increased band C levels by four- to fivefold over control in most experiments. In all, 47 out of the 108 genes tested were found to be active in regulating F508del-CFTR proteostasis (*Figure 2A–D*). Of these, 32 genes (when depleted) enhanced the levels of bands B and C by 1.5-fold to more than 10-fold over controls, while 15 genes decreased bands B and C by 20–80% of the control levels. We refer to these as anti-correction and pro-correction genes, respectively. Among these active genes, 30 were CORE genes and 17 were hubs in IPA networks. Notably, the correction that was induced by the depletion of many anti-correction genes was greater than that achieved by VX-809 (*Figure 2A*), or by the corrector drugs originally included in the study (see *Figure 2—figure supplement 2A*). This was in particular the case for a group of four poorly characterised ubiquitin ligases (RNF215, UBXO5, ASB8, FBXO7) that were not known to regulate F508del-CFTR proteostasis. RNF215 depletion increased the levels of bands C to over 10-fold the control levels (*Figure 2A*). Given these strong effects, RNF215 is a worthy candidate for further studies as a potential ERAD machinery component. Also notably, the depletion of many anti-correction genes not only enhanced the bands B and C but also markedly increased (to different extents) the band C/band B ratio (*Figure 2C*), suggesting that these genes affect the efficiency of export of F508del-CFTR protein from the ER and/or the stability of this protein after export. Given that siRNA treatments often have off-target effects, several controls were performed to ensure the specificity of the observed effects on proteostasis (see Methods section for details). It is also to be noted that the downregulation of anti-correction genes did not change the levels of F508del-CFTR mRNA (*Figure 2—figure supplement 2B*), suggesting that the observed effect is not due to increased F508del-CFTR synthesis, although these data cannot by themselves exclude an effect on translation. Further experiments confirm that the effect of the downregulation of these genes is mostly (if not completely) on F508del-CFTR degradation and folding/export (see below).

It might appear surprising to find both pro-correction and anti-correction genes within the downregulated CORE gene pool. However, these genes are, presumably, components of complex transcriptional modules whose role is to control cellular functions in a balanced manner. To this end, the concomitant operation of regulatory systems of opposite signs is probably necessary (*Hart and Alon, 2013*). These observations are therefore likely to be a reflection of the organization of the transcriptional programs that regulate proteostasis.

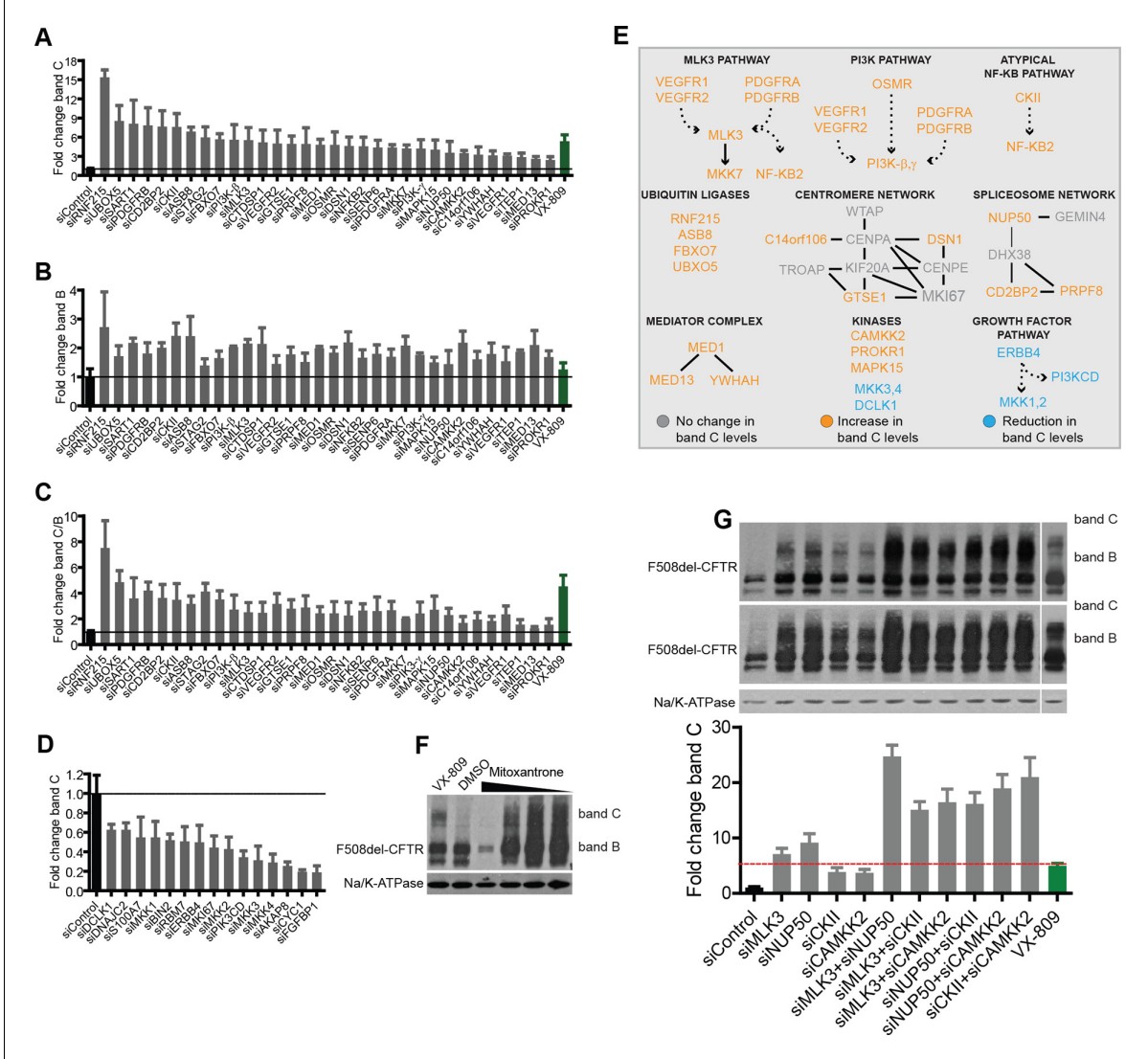

**Figure 2.** Validation of the selected CORE genes. (**A–D**) CFBE cells were treated with siRNAs targeting CORE genes and changes in F508del-CFTR proteostasis monitored by western blotting. The fold change in the levels of band C obtained by downregulating anti-correction (**A**) and pro-correction (**D**) genes and the fold change in levels of band B (**B**) and band C/band B ratio (**C**) after downregulation of the anti-correction genes are shown. The effects of negative control siRNAs (dashed line) and VX-809 (green) are indicated. (**E**) The validated CORE genes (blue – pro-correction hits, orange – anti-correction hits, and gray – no action) were assembled into coherent networks based on information from databases. Non-directional interactions denote protein-protein interaction, directional interactions represent phosphorylation cascades and dashed arrows indicate indirect connections through intermediaries. (**F**) Western blot of CFBE cells treated with mitoxantrone (2.5-20 μM for 48 hr), a potential corrector identified using downregulation of anti-corrector genes as selection criteria. Mitoxantrone increased the levels of both band C and band B. (**G**) Treatment of CFBE cells with the indicated combinations of siRNAs targeting CORE genes led to a synergistic increase in the band C levels. Changes in the levels of band C were quantitated from western blotting are presented as mean ± SEM (n > 3). The representative blots are shown in the insert, the upper panel corresponds to a blot with lower exposure where the differences in band C levels can be easily appreciated, while the bottom panel corresponds to a blot with higher exposure where a faint band C can be seen even under control siRNA treatment.

The following figure supplements are available for figure 2:

**Figure supplement 1.** Characterization of biochemical assay to monitor F508del-CFTR correction.

**Figure supplement 2.** Downregulation of CORE genes rescues F508del-CFTR more efficiently than the corrector drugs used originally, without altering the F508del-CFTR mRNA levels.

**Figure supplement 3.** The siRNAs efficiently reduce the transcript levels of their target genes.

Based on the above results, we sought to identify putative pathways/networks/groups (collectively networks) within the 47 *active* CORE gene pool, using literature data and pathway building tools (see *Figure 2E* for details). This resulted in several small potential networks (each comprising two to six connected elements), four of which were composed of signalling molecules and will be referred to by the name of their 'central' components: MLK3 (MAP3K11), PI3K, and CKII (with predominantly anti-correction activity), and ERBB4 (with pro-correction activity). A recent kinome-wide screening (published while this manuscript was being submitted) identified several kinases that regulate the rescue of F508del-CFTR (*Trzcinska-Daneluti et al., 2015*), with no overlap with the hits identified here (possibly due to the different functional assays and cell types used in that study versus ours). Other three of the networks shown in *Figure 2* comprised spliceosome, centromere and mediator complex components, and two were groups of ubiquitin-ligases and kinases.

## Proteostasis corrector drugs act in part by modulating the expression of CORE genes

We next sought to verify whether the effects of correctors on the CORE genes might explain the action of these drugs. We first analysed the frequency of the active CORE genes among the genes downregulated by the corrector drugs. The CORE genes were ~3-fold enriched in the signatures of correctors compared to those of other ~200 drugs taken at random from the MANTRA database (not shown). We next searched for MANTRA drugs that significantly downregulate the CORE genes (anti-correctors) using GSEA (specifically two-tailed symmetric GSEA as implemented in MANTRA; http://mantra.tigem.it/). The top 25 hits included three of the correctors that we had used for the FIT analysis. From the remaining 22, we selected eight drugs (based on availability) for testing in the correction assay. Among these, mitoxantrone was found to potently increase both band C and band B. (*Figure 2F*); in addition, among the top five hits was Vorinostat, an HDAC inhibitor that was shown to act as a corrector (*Hutt et al., 2010*). Thus, at least 20% (5 out of 22) of the short-listed drugs were correctors, while among a large number (>20) of randomly selected drugs none showed correction activity (not shown). These data suggest that the downregulation of CORE genes is a useful criterion to identify correctors. We then extended our analysis of the top hits by comparing the five active drugs with those that failed to correct and also by examining upregulated genes in their gene expression profiles. The correctors showed a high frequency (two- to threefold more than non-corrector drugs) of upregulation of the potent pro-corrector genes MKK1, MKK3 and FGFBP1, while the non-corrector drugs upregulated more frequently (two- to threefold more than corrector drugs) the anti-corrector genes NF-κB2 and MKK7. These results thus suggest that considering also the upregulation of CORE genes will help in further defining the search space for new correctors.

Altogether, the above data indicate that the drug-induced modulation of CORE genes is a significant component of the MOAs of corrector drugs. Thousands of gene signatures of drugs and perturbagens are being deposited in specialized databases (http://www.lincscloud.org/). These and a more extensive search for CORE genes will provide useful tools for a more refined bioinformatic identification of new correctors.

## Epistatic interactions between CORE pathways

As described earlier, an advantage of using this approach (deconvolution of drug MOA) to identify regulatory pathways is the possibility of discovering synergistic pathways. Thus, in order to explore the possible epistatic interactions between the CORE networks/pathways, siRNAs against selected targets were combined and tested on F508del-CFTR rescue. These candidates were chosen for their potential druggability and/or strong effects on correction. Strong synergistic interactions were observed between various combinations of siRNAs against CKII, CAMKK2, MLK3, and NUP50 (a spliceosomal network component) (*Figure 2G*), thus validating our choice of the approach. As a note of caution here, the efficacy of the combined siRNA treatments was more variable than that observed with single siRNAs. In our experience, this is because siRNAs in combinations are less effective than the individual siRNAs in depleting their target proteins, and a depletion threshold must be reached to achieve synergy. We conclude that, using the FIT technique and a series of bioinformatic and experimental filters, we have identified a set of synergistic molecular networks that show strong control over F508del-CFTR proteostasis.

## Delineation of the MLK3 and CAMKK2 signalling pathways regulating F508del-CFTR proteostasis

Next, we sought to define the composition and the role in correction of two representative CORE-networks, namely, the MLK3 and the CAMKK2 pathways. MLK3 (or MAP3K11) is part of a group of 14 MAP3 kinases that act through cascades of MAP2K and MAPK enzymes. MLK3 can be activated by various PM receptors, including the TNF-α, TGF-β, VEGF, and PDGF receptors, through at least two MAP4Ks (haematopoietic progenitor kinase [HPK]1 and germinal centre kinase [GCK]) and glycogen synthase kinase (GSK)3β, or via the CDC42/Rac family (summarised in [*Schachter et al., 2006*]). MLK3 can also be activated by stress, e.g., oxidative stress (*Lee et al., 2014*) (i.e., it is a Stress Activated Protein Kinase, or SAPK) and it can, in turn, trigger three main kinases: p38 MAPK, c-Jun N-terminal kinase (JNK), and extracellular signal regulated kinase (ERK), depending on cell type and conditions, through the intermediate kinases MKK3/6, MKK4/7 and MKK1/2, respectively (*Schachter et al., 2006*). MLK3 is also known to be an upstream activator of NF-kB (*Hehner et al., 2000*). We thus sought to determine which components of the MLK3 pathway have roles in F508del-CFTR correction.

The VEGF and PDGF receptors, MKK7, and NF-κB2, like MLK3, appear to be components of the correction-relevant branch of the MLK3 pathway, as indicated by the screening data in *Figure 2A*. Among the components upstream of MLK3, we found TGF receptors, CDC42, Rac2, and HPK1 to be active in correction (i.e. their depletion induced correction) (*Figure 3A*). Within the cascade downstream of MLK3, MKK7 (*Figure 2A*) and further downstream, JNK2 (*Figure 3B*) were active components (JNK2 is highly expressed in bronchial epithelial cells [http://biogps.org]). The p38 MAPK, also downstream of MLK3 (through MAP2K3 and MAP2K6) was inactive in CFBE but moderately active in HeLa cells, indicating some cell type-dependent specificity in the effects of these kinases (*Figure 3—figure supplement 1A*). Thus, altogether, suppression of the MKK7-JNK2 branch of the MLK3 pathway induced F508del-CFTR correction. Conversely, when the activity of the MLK3 pathway was enhanced by transfection of the MLK3 activator CDC42 or MKK7 or JNK2 into the CFBE cells, the levels of both bands B and C dropped markedly (*Figure 3C*), confirming that the MLK3 pathway has a tonic negative effect on the proteostasis of F508del-CFTR.

In sum, as shown in *Figure 3D*, a signal regulating F508del-CFTR proteostasis flows from the ligands and receptors upstream of MLK3, through HPK1 and CDC42/Rac2, to impinge on MLK3 and is then passed on through the JNK2 arm. NF-κB2 is also a probable downstream component of this proteostasis regulatory pathway. We also tested (again by siRNA silencing) seven other MAP3Ks (including TAK1/MAP3K7, see below) that can activate JNK or p38, for their effect on F508del-CFTR proteostasis. They had no effect (not shown). This highlights the remarkable specificity of MLK3 in the regulation of proteostasis, possibly due to spatial/temporal compartmentalization of the MAPK networks (*Engstrom et al., 2010*).

A similar series of experiments were performed to characterise the CAMKK2 cascade in F508del-CFTR correction. The results are reported in detail in *Figure 3—figure supplement 1B,C* (see also *Figure 3E*), and indicate that the CAMKK2 pathway has negative effects on F508del-CFTR proteostasis similar to those found for the MLK3 pathway.

## The MLK3 pathway exerts complex regulatory effects on F508del-CFTR proteostasis

The increase in the levels of band B induced by inhibition of the MLK3 pathway might be due to increased synthesis or due to decreased degradation of F508del-CFTR. Downregulation of MLK3 did not increase the CFTR mRNA levels (*Figure 2—figure supplement 2B*), speaking against the former possibility, although an effect of this pathway on the translational efficiency cannot be excluded. We then examined the degradation of band B using both a cycloheximide (CHX) chase and a radioactive pulse-chase assay. Downregulation of the MLK3 pathway markedly slowed the degradation of band B when measured by CHX chase assay (*Figure 4A,B*) and similar effects were obtained with the radioactive pulse-chase method (*Figure 4—figure supplement 1A,B*). We also examined the effects of enhancing the activity of MLK3 pathway by overexpressing CDC42 or MKK7 or JNK2: under these conditions the rate of degradation of band B increased twofold (*Figure 4C,D*; see also [*Ferru-Clement et al., 2015*]). Notably, the ubiquitin-proteasome system itself was not detectably affected by the modulation of the MLK3 pathway activity, as judged by the lack of effects on both the

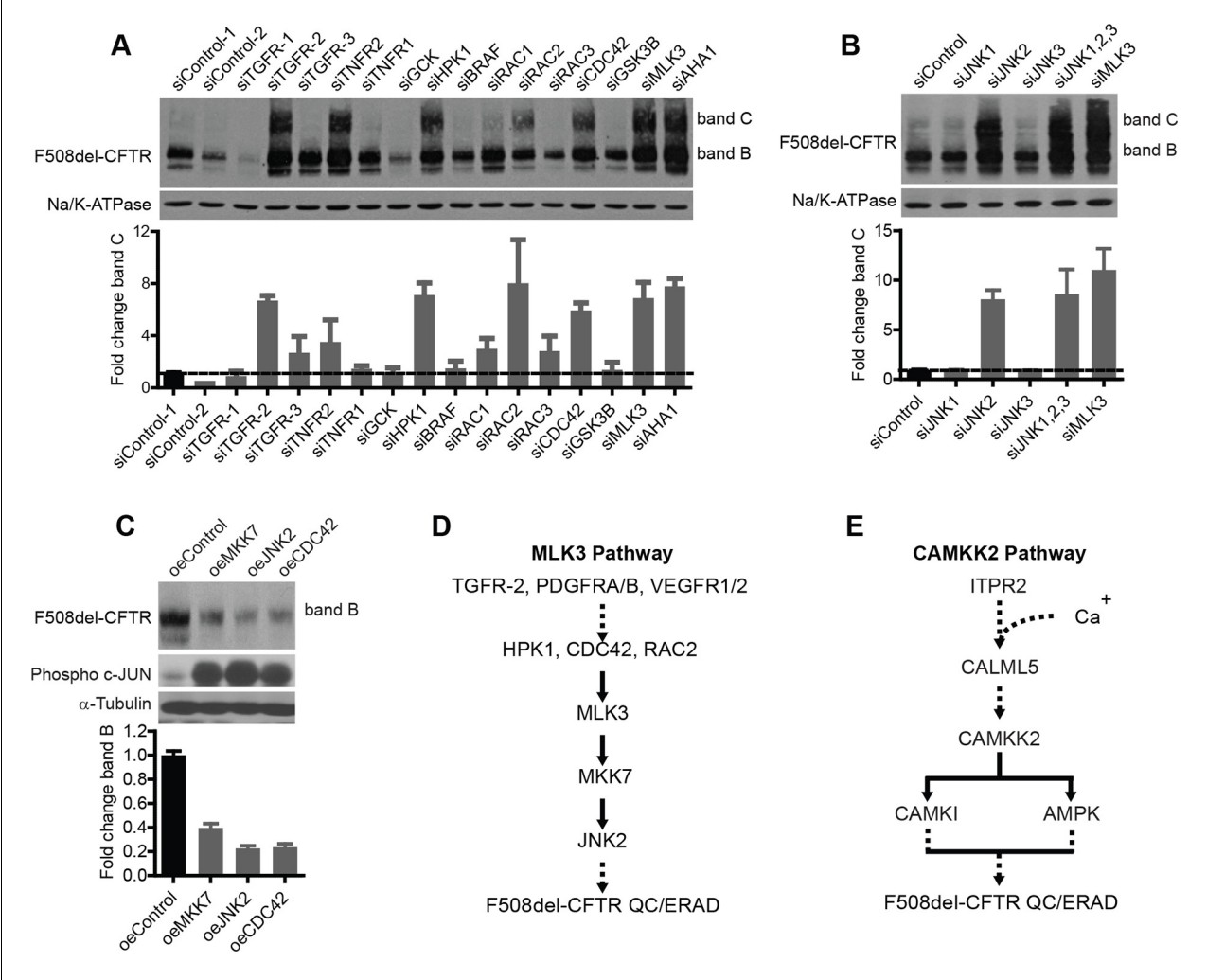

**Figure 3.** Delineation of the MLK3 pathway branch that controls F508del-CFTR proteostasis. (**A**) CFBE cells were treated with indicated siRNAs targeting the upstream activators of MLK3 and their effect on F508del-CFTR proteostasis monitored by western blotting. The fold change in band C levels is shown as mean ± SEM (n > 3). Reduction in TGF receptor, HPK, CDC42 and RAC2 levels rescued F508del-CFTR from ERQC. The rescue obtained with TNFR2 siRNA was quite variable and thus was not considered further. (**B**) JNK isoforms were tested for their effect on F508del-CFTR proteostasis after siRNA-mediated downregulation of their levels. Downregulation of JNK2 leads to efficient rescue of F508del-CFTR that is comparable to that obtained with MLK3. The fold change in band C levels is presented as mean ± SEM (n > 3) with western blot in the insert. (**C**) CFBE cells were transfected with activators of the MLK3 pathway to study their effect on F508del-CFTR proteostasis. The fold change in the band B levels is shown as mean ± SEM (n > 3) with western blot in the insert. All of them reduced the levels of both band B and band C (not shown) of F508del-CFTR. The corresponding increase in the levels of phospho-c-jun indicates an increase activation of the MLK3 pathway activity. (**D, E**) Schematic representation of the proposed MLK3 (**D**) and CAMKK2 (**E**) pathways that regulate F508del-CFTR proteostasis. The directional interactions proposed between the components of the pathways are based on published literature.

The following figure supplement is available for figure 3:

**Figure supplement 1.** Delineation of the MLK3 and CAMKK2 pathway branches that regulate F508del-CFTR proteostasis.

proteasome sensor-ZsProsensor-1 (*Figure 4—figure supplement 1C*) and the accumulation of poly-ubiquitinated proteins (*Figure 4—figure supplement 1D*). Thus, the MLK3 pathway appears to regulate the ERQC/ERAD of F508del-CFTR at a step prior to proteasomal digestion.

In addition, silencing of the MLK3 pathway (and of several CORE genes) increased also the band C/band B ratio (see *Figure 2C*). This is not explained by reduced ERAD alone and suggested that the MLK3 pathway might have additional effects on the folding/ export of F508del-CFTR, or on the stability of band C at the PM (or both). The trypsin susceptibility assay to assess the folding status of

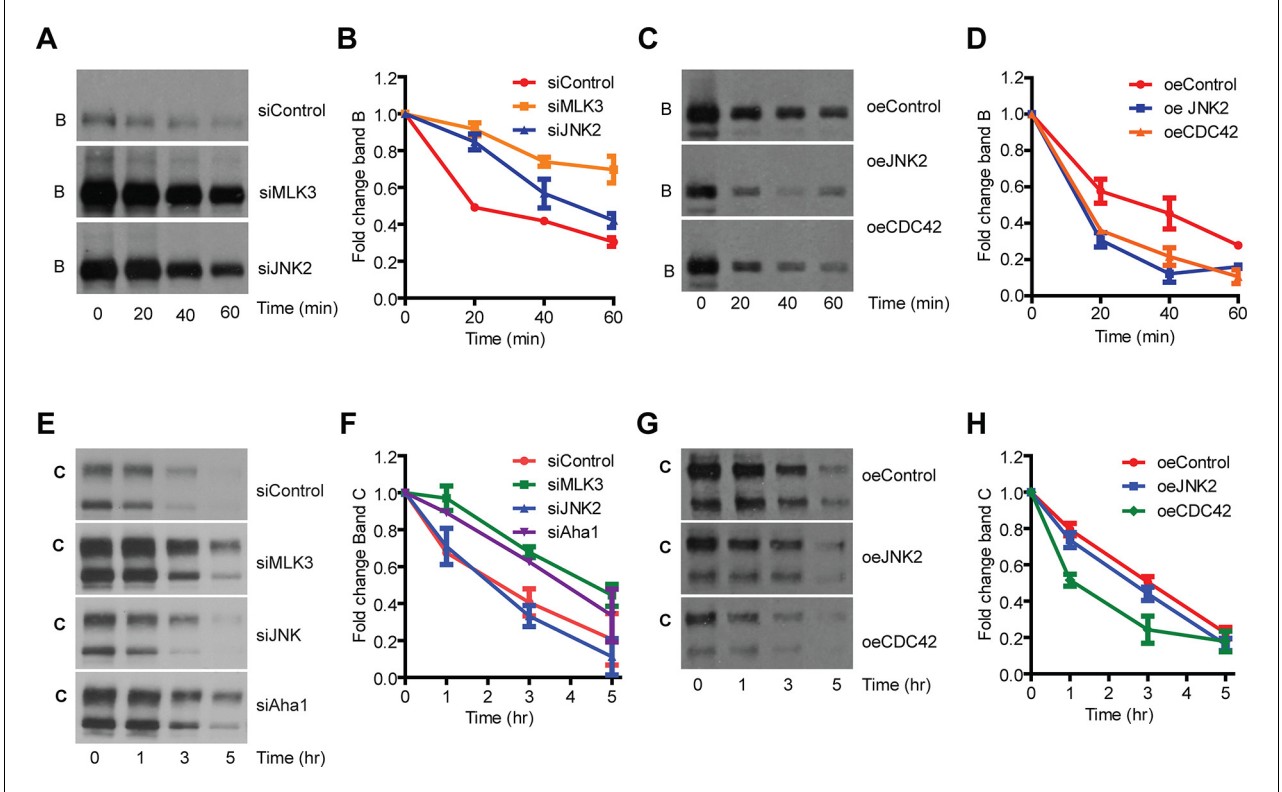

**Figure 4.** MLK3 pathway regulates the degradation of F508del-CFTR. (**A, B**) CFBE cells pretreated with siRNAs were treated with CHX (50 μg/mL) for indicated times and the levels of band B of F508del-CFTR was monitored (**A**). The levels were quantitated and represented in (**B**) as mean ± SEM (n > 3). Downregulation of MLK3 or JNK2 reduced the kinetics of reduction of band B of F508del-CFTR. (**C, D**). CHX chase assay (see above) after overexpression of the activators of MLK3 pathway. The activation of MLK3 pathway increases the rate of degradation of band B (**C**). Quantitation of the blot is shown in (**D**) as mean ± SEM (n > 3). (**E, F**) CFBE cells were treated with indicated siRNAs followed by incubation at 26°C for 6 hr followed by shift to 37°C for the indicated time periods. The changes in band C levels were monitored as measure of PQC (**C**). See (**F**) for quantitation of band C levels represented as mean ± SEM (n > 3). (**G, H**). PQC assay (see above) after overexpression of CDC42 or JNK2 shows an increased rate of degradation of band C (**G**) upon CDC42 overexpression. JNK2 overexpression has no effect on the PQC of F508del-CFTR. The blots were quantified and presented in (**H**) as mean ± SEM (n > 3).

The following figure supplement is available for figure 4:

**Figure supplement 1.** Characterization of the mode of action of the MLK3 pathway on F508del-CFTR proteostasis.

F508del-CFTR and an assay for protein transport out of the ER using vesicular stomatitis virus G protein (VSVG), a classical probe to study secretory trafficking, ruled out large effects of the MLK3 pathway on F508del-CFTR folding or on the general ER-export machinery (*Figure 4—figure supplement 1E,F*). Nevertheless, we note that these assays (trypsin susceptibility or VSVG export) are limited in their scope and do not capture the wide spectrum of subtle regulations that can influence the outcome of proteostasis (see below for a discussion of the effect of the MLK3 pathway on folding/export). We next tested the effect of MLK3 on the stability of F508del-CFTR at the PM. We depleted MLK3 and exposed the cells to low temperature (26°C), to accumulate F508del-CFTR at the cell surface, and then shifted the cells back to 37°C, a temperature at which the F508del-CFTR at the PM is subjected to accelerated ubiquitination and degradation (*Okiyoneda et al., 2010*). Under these conditions, the depletion of MLK3 slowed the degradation rate of band C, increasing the $t_{1/2}$ from ~2 to ~4 hr (*Figure 4E,F*), whereas overexpression of CDC42 to activate MLK3 enhanced the band C degradation rate (*Figure 4G,H*). These data suggest that also the peripheral QC of F508del-CFTR is regulated by MLK3. In contrast, the knockdown of JNK2 (or its overexpression) did not change the degradation kinetics of band C, although it increased the band C/band B ratio (not shown), suggesting that JNK2 may have additional effects, for instance on the folding and/or export of F508del-

CFTR (see above and Discussion). Similar effects on F508del-CFTR folding seem likely to be induced also by many of the CORE genes whose depletion greatly increases the band C/band B ratio, in some cases up to 4-fold over control levels (*Figure 2C*).

In conclusion here, depletion of the receptor and stress-activated MLK3 signalling pathway markedly inhibited both the ER-associated and peripheral degradation processes of F508del-CFTR while possibly at the same time increasing the efficiency of F508del-CFTR folding/ER export. Consequently, inhibition of this pathway results in large increases in the levels of the Golgi-processed mature form of F508del-CFTR.

## ROS and the CF modifiers TNF-α, TGF-β enhance F508del-CFTR degradation in an MLK3–dependent fashion

We next examined the effects on F508del-CFTR proteostasis of agents known to activate MLK3 such as TNF-α, TGF-β (*Schachter et al., 2006*) and reactive oxygen species (ROS) (*Lee et al., 2014*). TNF-α and TGF-β have been proposed to be genetic modifiers of CF (*Cutting, 2010*), and ROS have been reported to be enhanced in CF cells (*Luciani et al., 2010*) and to be massively produced by neutrophils during the inflammatory reactions that are common in CF patients (*Witko-Sarsat et al., 1995*). We treated CFBE cells with TNF-α, TGF-β or $H_2O_2$ (to increase ROS), and monitored the effects on F508del-CFTR. The effects of $H_2O_2$ at non-toxic concentrations were dramatic, with a marked drop of the F508del-CFTR levels within 30 min (*Figure 5A,B*). Also TNF-α and TGF-β induced rapid, although less complete (50%) decreases in levels of F508del-CFTR (*Figure 5C,D*). Under these conditions, the reduction in F508del-CFTR levels was completely abolished by MLK3 downregulation (*Figure 5A–D*), confirming the crucial role of MLK3 pathway in F508del-CFTR QC/degradation.

These results, and in particular the effects of $H_2O_2$, provide evidence for the existence of a rapid and potent mechanisms of protein degradation that depend on the MLK3 pathway and act on F508del-CFTR (and presumably on other misfolded mutant proteins). These regulatory mechanisms might have pathological relevance, as discussed below.

## Chemical inhibitors of the MLK3 pathway act as CFTR correctors and potently synergize with the pharmacochaperone VX-809

We next tested the effect of selected kinase inhibitors on F508del-CFTR proteostasis in CFBE cells. A well-known characteristic of the kinase inhibitors is their promiscuity. In our experience, inhibitors that nominally target the same kinase can cause divergent effects on correction (see below), most likely because they target other kinases with different or competing effects. We sought to overcome this difficulty by selecting kinase inhibitors with different structures and modes of action, and by using information from the KINOMEscan library (http://lincs.hms.harvard.edu/data/kinomescan/). For JNK, we tested a set of 10 reported JNK inhibitors (JNKi), three of which led to robust increases in the levels of band B and band C (*Figure 6A–D*; JNKi II, JNKi IX, JNKi XI) at concentrations that were required for JNK inhibition in CFBE cells (*Figure 6—figure supplement 1A*). These JNK inhibitors have different chemical structures; moreover, while JNKi II and JNKi IX are ATP-competitive inhibitors of JNK, JNKi XI is an inhibitor of substrate/scaffold binding to JNK. Therefore, they appear to be reliable tools to correct F508del-CFTR by targeting JNK. A previously proposed MLK3 inhibitor (K252a) had no clear effects on correction, perhaps because it inhibits MLK3 weakly and has diverging effects on other kinases (see http://www.kinase-screen.mrc.ac.uk/screening-compounds/345892). We thus searched the KINOMEscan library for a molecule that had a suitable inhibitory pattern on the MLK3 pathway. (5Z)-7-oxozeaenol (hereafter referred to as oxozeaenol) (*Ninomiya-Tsuji et al., 2003*) potently inhibits the MLK3 pathway members VEGF and PDGF receptor kinases and (less potently) MLK3 itself and MKK7, as well as, more weakly, a few kinases with antagonistic effects on correction (http://lincs.hms.harvard.edu/db/datasets/20211/). Oxozeaenol markedly increased the bands B and C of F508del-CFTR (*Figure 6A–C*). This drug had been identified as a corrector in an earlier screening study, and proposed to have F508del-CFTR corrector properties as an inhibitor of TAK1 (MAP3K7) (*Trzcinska-Daneluti et al., 2012*). However, the downregulation of TAK1 itself had no effect on correction (*Figure 6—figure supplement 1B*). Although the pharmacological inhibition and the siRNA knockdown of kinases can sometimes have distinct consequences, these data suggest that oxozeaenol likely acts through MLK3 pathway kinases to affect proteostasis rather that through

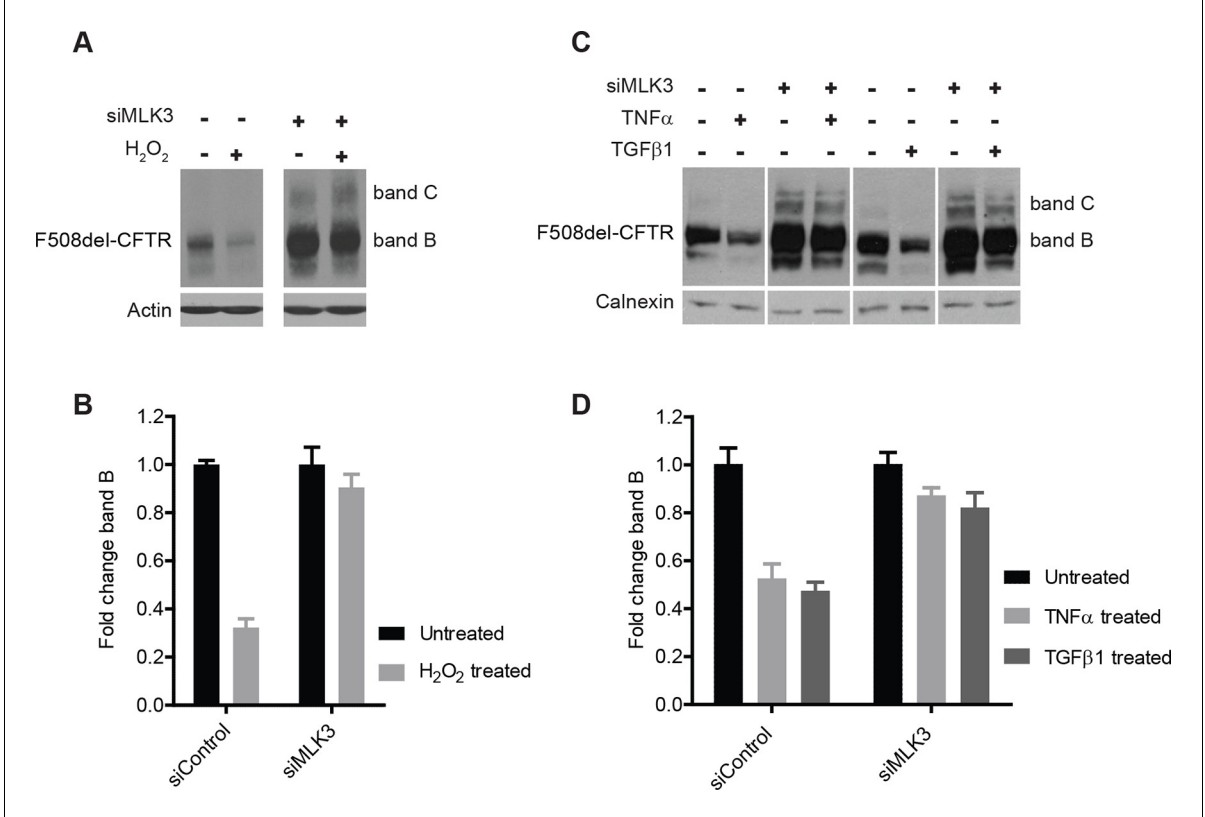

**Figure 5.** ROS and inflammatory cytokines control F508del-CFTR proteostasis via the MLK3 pathway. (**A**) CFBE cells (pre-treated with MLK3 siRNA) were treated with 1 mM H$_2$O$_2$ for 30 min and levels of band B monitored by western blotting. Control cells show drastic reduction in F508del-CFTR levels upon treatment with H$_2$O$_2$ that is prevented by the downregulation of MLK3. (**B**) Quantitation of band B levels from (**A**) represented as mean ± SEM (n > 3). (**C**) CFBE cells (pre-treated with MLK3 siRNA) were treated with 50 ng/mL of TNF-α or TGF-β for 15 min and levels of band B monitored by western blotting. MLK3 downregulation prevents the decrease in band B levels brought about by treatment with the cytokines. (**D**) Quantitation of the band B levels from (**C**) represented as mean ± SEM (n > 3).

TAK1. In line with this notion, the corrective effects of oxozeaenol were not additive with MLK3 knockdown (*Figure 6—figure supplement 1C*) and were accompanied by a reduction in phospho c-jun levels (c-jun phosphorylation is diagnostic of JNK activity) (*Figure 6—figure supplement 1D*).

In sum, selected chemical blockers of the MLK3 (and CAMKK2; *Figure 6—figure supplement 1E*) pathway potently increase the levels of band B and band C as well as the band C/band B ratio (*Figure 6A–D*). The level of correction obtained with these inhibitors is higher than the effects of the corrector compounds from which the pathways were deduced (*Figure 2—figure supplement 2A*), and similar to, or higher than, the effects of VX-809.

We next considered that, while inhibitors of MLK3 pathway lead to large increases in the ER-localised band B, the pharmacochaperone VX-809 increases F508del-CFTR band C levels with only a limited effect on band B (*Figure 6B–D*), presumably because it primarily enhances F508del-CFTR folding. This suggested that if the large pool of band B protein accumulating in the ER following the inhibition of the MLK3 pathway is in a foldable state, VX-809 should act on such pool to enhance its folding, and greatly increase the generation of the band C mature protein. Indeed, when we added both MLK3 pathway inhibitors and VX-809, there was potent synergy between them (*Figure 6E,F*) with increases in levels of band C that were over 20-fold the basal band C level and fourfold over those obtained with VX-809 alone. Given the promising results of VX-809 in combination therapies in recent clinical trials (*Wainwright et al., 2015*) and in experimental settings (*Okiyoneda et al., 2013*; *Phuan et al., 2014*), the observed additive/synergistic effects of VX-809 combined with MLK3 pathway inhibitors might provide a potential therapeutic opportunity.

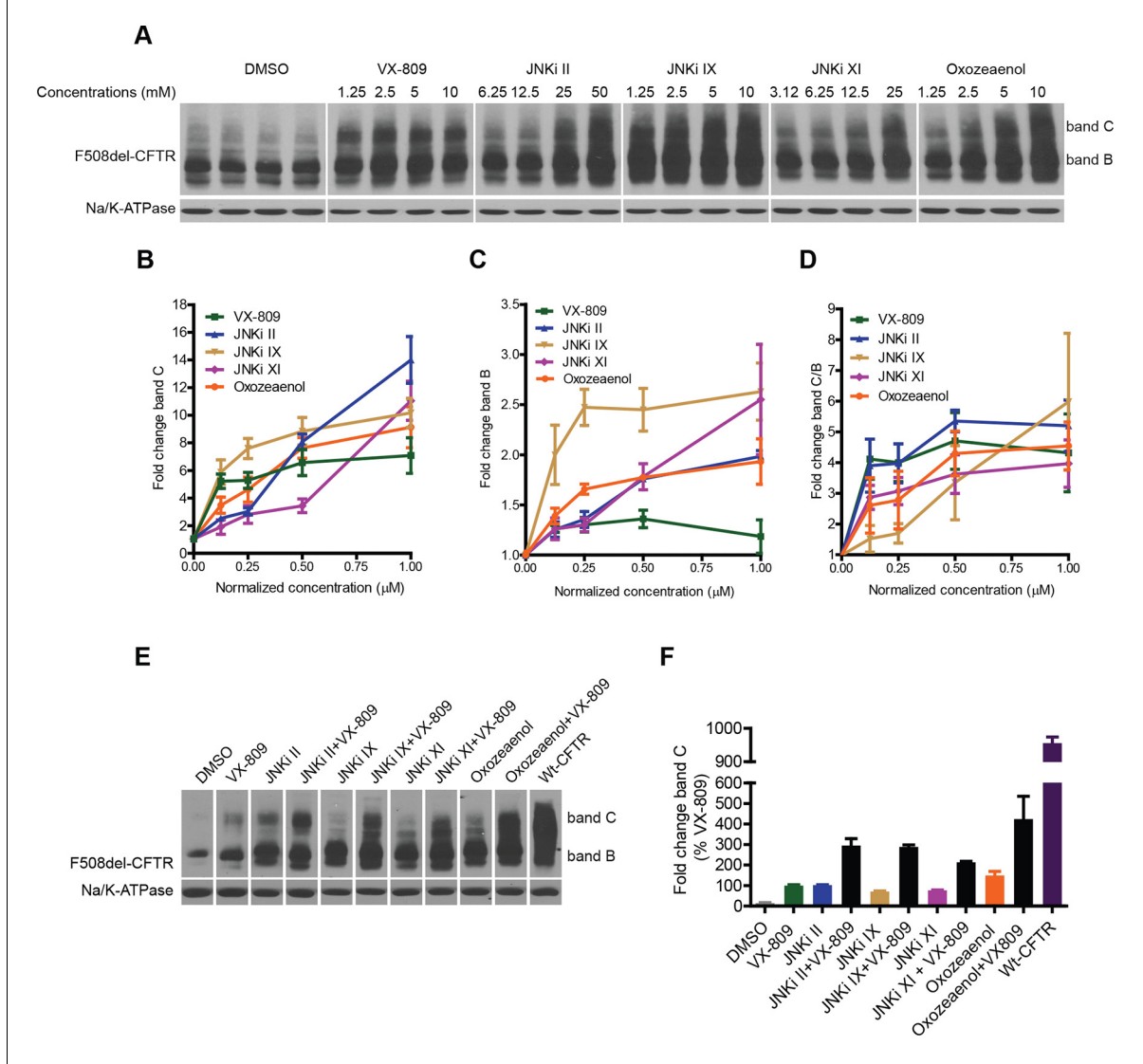

**Figure 6.** Inhibitors of the MLK3 pathway rescue F508del-CFTR. (**A–D**) CFBE cells were treated with the indicated inhibitors of the MLK3 pathway or VX-809 for 48 h, and the rescue of F508del-CFTR from ERQC was monitored (**A**). MLK3 pathway inhibitors rescue F508del-CFTR to levels comparable to or more than that achieved by VX-809. Changes in the levels of band C (**B**), band B (**C**) and the band C/band B ratio (**D**) were quantitated and shown mean ± SEM (n > 3). Normalized concentration (abscissa in panels **B–D**) refers to concentration (VX-809, JNKi IX and Oxozeaenol [1.25, 2.5, 5, 10 µM], JNKi II [6.25,12.5, 25, 50 µM], JNKi XI [3.12, 6.25,12.5, 25 µM]) values that were normalized to the maximum used concentrations of the respective drugs. Refer panel A for concentrations (µM) of the drugs used. (**E**) CFBE cells were treated with inhibitors of the MLK3 pathway and/or VX-809 (5 µM) for 48 hr and changes in band C levels monitored. The concentrations of the MLK3 pathway inhibitors used were: JNKi II (12.5 µM), JNKi IX (5 µM), JNKi XI (25 µM) and oxozeaenol (5 µM). Wild-type CFTR (wt-CFTR) was used as a control. (**F**) Quantitation of band C levels from (**E**) normalized to the levels of band C after VX-809 treatment are shown as mean ± SEM (n > 3). The results show that synergy obtained between the MLK3 pathway inhibitors and VX-809 brings the levels of band C to ~40% of the wild-type levels.

The following figure supplements are available for figure 6:

**Figure supplement 1.** Small-molecule inhibitors of the MLK3 pathway rescue F508del-CFTR and other structurally related mutant proteins from degradation.

**Figure supplement 2.** Small molecule inhibitors of MLK3 pathway rescue channel function of F508del-CFTR.

## The MLK3 pathway exerts selective effects on the proteostasis of F508del-CFTR and of structurally related mutant proteins

We next examined the effects of the MLK3 pathway inhibition on the proteostasis of other conformational disease mutants. We transfected CFBE (and HeLa) cells with different conformational mutants (i.e., Sodium-chloride symporter [NCC, R948X mutant]; P-glycoprotein, [P-gp, G268V and DY490 mutants]; human *Ether-à-go-go*-Related Gene [hERG, G601S mutant]; Wilson's disease associated protein [ATP7B, H1069Q and R778L mutants]) and then treated cells with JNKi II. The effects on their proteostasis was monitored by assessing changes in their glycosylation patterns (NCC, P-gp, hERG mutants) or in their intracellular movement from the ER to the Golgi complex (ATP7B mutants). JNKi II rescued some of these mutants (P-gp DY490 and ATP7B mutants), while it had no effects or had 'negative' effects, on others (*Figure 6—figure supplement 1F*). The effects on ATP7B were large and led to almost complete correction, as we report elsewhere (*Chesi et al., 2015*). We also tested the effect of JNKi II on other endogenous non-mutant proteins (apart from Na$^+$/K$^+$-ATPase that was a reference in many of our treatments) including, E-cadherin, IGF1R-β and EGFR (*Figure 6* and *Figure 6—figure supplement 1G*) and found no effect on their proteostasis. Both P-glycoprotein and ATP7B, like CFTR, have two groups of transmembrane domains with an interconnecting nucleotide-binding domain. Moreover, the mutations tested (DY490 and H1069Q) are located in the nucleotide-binding domains of these proteins, and result from either a loss or substitution of aromatic amino acids, as for F508del-CFTR. These similarities suggest that a common proteostatic machinery might be involved in the processing of these mutants and can be targeted by the MLK3 pathway in a selective fashion.

## Discussion

In this study, we have developed a bioinformatic method based on the fuzzy intersection of drug transcriptomes (FIT) that reveals the transcriptional components of the MOAs of proteostasis correctors. Using this method, we have uncovered a set of correction-relevant genes (CORE genes), several of which belong to signalling networks that potently and selectively regulate the proteostasis of F508del-CFTR and of structurally related protein mutants.

### Physio-pathological significance of the CORE signalling networks

Based on literature data, interaction databases and our own experimental findings, the correction-relevant components we identified can be organised into five signalling cascades, which, for brevity, we refer to as the MLK3, CAMKK2, PI3K, CKII, and ERBB4 networks. Other networks were made up of constituents of the spliceosome, centromere and mediator (transcriptional) complexes, or of ubiquitin ligases.

The physiological role of these CORE signalling systems might be to regulate the stringency of the QC and degradation processes. Most of the CORE pathways enhance the efficiency of QC and degradation. This is the case of the MLK3 pathway, which can be activated by selected cytokines and by cellular stresses. The ERBB4 pathway, in contrast, is activated under growth conditions, and appears to have the effect of suppressing the QC and degradation processes. It may be speculated that cells under stress need to reduce the toxic burden of certain classes of unfolded proteins to survive, while growing cells might need to 'tolerate' higher levels of folding/unfolded proteins to proliferate, and that the CORE pathways regulate the proteostasis machinery according to needs. A further possibility is that at least part of the CORE pathways might function as part of an internal control system (*Cancino et al., 2014*; *Luini et al., 2014*) that is activated by the presence of unfolded/misfolded proteins. Interestingly in this regard, MLK3 interacts directly with (and might be activated by) HSP90 (*Zhang et al., 2004*), a component of the F508del-CFTR folding and QC machinery.

Under pathological conditions such as cystic fibrosis and similar diseases, the activity of MLK3 and other core pathways can become deleterious, as it enhances the degradation of protein mutants that retain the potential to function (such as F508del-CFTR). Also importantly, they can be hyperactivated under pathological conditions, leading to vicious circles. For example, large amounts of ROS are produced by neutrophils in the inflamed lungs of CF patients (*Witko-Sarsat et al., 1995*), and elevated serum VEGF are detected in some CF patients (*McColley et al., 2000*). Both of these

molecules act via the MLK3 pathway to enhance the degradation of F508del-CFTR, and in particular the ROS do so with striking efficacy and speed (*Figure 5A,B*). These effects most probably result in lowering F508del-CFTR to levels below those that would be determined by the primary folding defect, which might be harmful because even low residual levels of F508del-CFTR may help to improve the CF phenotype in the long term (*Amaral, 2005*). Blocking the MLK3 pathway is thus probably important to stop maladaptive processes that can adversely affect therapeutic efforts. Similar considerations apply to the CAMKK2 and other CORE-derived pathways.

## Mechanism of action of the MLK3 signalling network

The ER quality control relies on chaperones such as HSP90 and HSC70 that are also involved in folding and can switch between folding and quality control/degradation roles depending on their dwell-time on the folding client proteins (*Zhang et al., 2013*). The simplest interpretation of the data is therefore that inhibition of the MLK3 pathway regulates this folding/degradation switch by impairing the entry of F508del-CFTR into the degradation pathway and giving the mutant more time to fold and exit the ER. Although MLK3 does not measurably affect the folding of F508del-CFTR as measured by the trypsin susceptibility assay, it cannot be excluded, however, that MLK3 (and other CORE genes) might exert subtle direct actions on the folding and/or ER export mechanisms. This is supported by the strong effects of some of the CORE pathways on the band C/band B ratio, and also by the observation that the inhibition of MLK3 markedly stimulates the efficiency of export of a mutant of ATP7B (similar in structure to CFTR) from the ER (*Chesi et al., 2015*).

At the molecular level, the mechanisms underlying these rescue effects remain unclear. Some initial insight might come from our observation that the phosphoprotein HOP co-precipitates much less efficiently with F508del-CFTR in cells treated with JNK inhibitors than in control cells (*Figure 4—figure supplement 1G*), while its amount remains unchanged. HOP serves as a link between HSC70 and HSP90 in the F508del-CFTR folding process, and its depletion induces rescue of F508del-CFTR (*Marozkina et al., 2010*), possibly by acting on the folding/ERQC switch discussed above. Thus, a reduced interaction of HOP with F508del-CFTR–associated QC/folding complex might be one of the possible modes of action of MLK3 on F508del-CFTR rescue. This might be an example of how corrector drugs may act by modulating the 'activity', rather than the levels, of relevant machinery proteins. A complete analysis of the effects of the MLK3 pathway on the interactions and posttranslational modifications of the ERQC/ERAD machinery components remains a task for future work.

## Relevance of the CORE signalling networks for the development of clinically useful proteostasis correctors

As already noted, this study does not aim to generate efficient correctors ready for clinical use. Rather, it aims to elucidate the regulatory signalling networks that control a central element of the disease viz. the proteostasis machinery acting on F508del-CFTR, with a view to providing a rational basis to identify relevant pharmacological targets and, in the long run, more effective F508del-CFTR correctors.

This goal appears realistic, because signalling cascades are eminently druggable (the majority of the known drug targets are signalling components [*Imming et al., 2006*]), and an enormous repertoire of drugs directed at kinases and other related molecules has been developed by the pharmaceutical industry for the therapy of major diseases. For instance, over 120 inhibitors against the correction-related kinases identified in this study (not shown) are currently in clinical trial. Moreover, as shown for the case of oxozeaenol (*Figure 6A–D*), suitable kinase inhibitors can be selected in a rational fashion by matching the list of CORE kinases with the kinase inhibitory patterns of the many available drugs of this class, according to polypharmacology principles (*Aggarwal et al., 2007*). It is thus quite possible that some of these drugs may be repositioned for the CF therapy. In addition, the CORE ubiquitin ligases, particularly RNF215, are also attractive targets in view of their potent effect on F508del-CFTR correction (*Figure 2A*). Although the technology for developing ubiquitin ligase inhibitors is still in its early stages, robust progress is being made in this direction (*Goldenberg et al., 2010*).

A further consideration is that the inhibitors of the CORE pathways show corrective effects that are (partially) selective for F508del-CFTR (and structurally related mutants) (see *Figure 6—figure*

*supplement 1F*); and that these effects are complementary and synergic with those of the pharma-cochaperone VX-809. Since these synergies lead to levels of correction that are several-fold higher than those achieved by VX-809 alone, it is possible that they result in combination therapies of clinical interest. Also of note is that the MOA-based approach used here can be exploited further in the future to identify more CORE pathways as well as more effective and specific correctors.

A remaining obstacle is the need to combine the correction of proteostasis, which is the focus of this manuscript, with the restoration of the activity of the channel. The restitution of chloride channel conductance at the PM that we observe with the inhibitors of MLK3 pathway does not always match up to the level of rescue obtained on proteostasis alone (see also *Figure 6—figure supplement 2*). Notably, a similar quantitative discrepancy between the effect on proteostasis and the effect on chloride channel activity has been observed before (see [*Hutt et al., 2010*]). Regulation of the chloride channel activity is a complex phenomenon and is determined by several factors including the input from signaling pathways (for e.g. PKA pathway and others), and the presence (or absence) of regulators such as NHERF and cytoskeletal components. Further studies that reveal the regulatory network controlling the activity of the channel at the PM will complement our study on proteostasis and will help in the rational design of pharmaceutical approaches.

In addition, a key requirement for translating our findings towards clinical treatments is the conservation of the CORE pathways in bronchial epithelial cells in situ. We have observed fundamentally similar roles of the CORE networks across several human and mammalian cell lines, both under polarized and non-polarized conditions, suggesting that these networks are well conserved. Moreover, JNK has been reported to be hyperactive in the lungs of a mice model of CF (*Grassme et al., 2014*), as is p38 MAPK (also activated by MLK3) in the lungs of CF patients (*Berube et al., 2010*), indicating that a SAPK pathway is activated under these conditions. Also notably, the MLK3 pathway inhibitor oxozeaenol has been reported to be effective in correcting the F508del-CFTR proteostasis defect in the primary human bronchial epithelial cells (*Trzcinska-Daneluti et al., 2012*). These observations, together with the fact that the CF genetic modifiers TNF-α and TGF-β potently affect F508del-CFTR proteostasis, support the notion that a regulatory network similar to that uncovered in CFBE cells operates on the proteostasis machinery in bronchial epithelial cells in CF patients.

In sum, this study builds on previous screening studies and on the accumulated knowledge about the F508del-CFTR proteostasis machinery (*Balch et al., 2011*; *Farinha et al., 2013b*; *Lukacs and Verkman, 2012*; *Turnbull et al., 2007*) to identify signalling pathways acting on F508del-CFTR proteostasis. This provides new insight into the physiopathology of F508del-CFTR and opens new possibilities to pharmacologically correct the folding and trafficking defect of this mutant protein. To establish the efficacy of these interventions in human bronchial epithelia and relevant animal models (*Yan et al., 2015*) will be the next stage towards the rational development of effective F508del-CFTR proteostasis regulators for patients with CF.

## Materials and methods

### Cell culture, antibodies, plasmids and transfection

Cell lines used in this study are characterized and reported CF model systems. These include CFBE cells stably expressing wild type CFTR or F508del-CFTR (*Bebok et al., 2005*) and stably expressing halide sensitive YFP (*Pedemonte et al., 2005*) and HeLa cells stably expressing HA-tagged F508del-CFTR (*Okiyoneda et al., 2010*) that were obtained after material transfer agreements from the respective laboratories. These cell lines are not commercial and their STR status is unknown. Mycoplasma contamination was not observed in the cell cultures. CFBE cells were cultured in Minimal Essential Medium supplemented with 10% foetal bovine serum, non-essential amino acids, glutamine, penicillin/streptomycin and 2 μg/ml puromycin. This media additionally supplemented with 50 μg/ml G418 was used for the CFBE-YFP cells. HeLa cells were cultured in Dulbecco's modified Eagle's medium (DMEM) supplemented with 10% foetal bovine serum, glutamine, penicillin/streptomycin and 1 μg/ml puromycin.

The antibodies used were: anti-phospho-c-jun, EGFR, IGF1R-β (Cell Signaling Technology, Danvers, MA), E-cadherin (Abcam, UK), monoclonal anti-HA, anti-actin and anti-tubulin (Sigma, St. Louis, MO), rat anti-CFTR (3G11; CFTR Folding Consortium), mouse monoclonal anti-

CFTR (M3A7), HRP-conjugated anti-mouse, rabbit and rat IgG (Merck Millipore, Germany) and anti-Na/KATPase α1 (Thermo Fisher Scientific, Waltham, MA).

The plasmids used were: JNK2 (pCDNA3 Flag MKK7B2Jnk2a2; Addgene plasmid #19727) and MKK7 (pCDNA3 Flag MKK7b1; Addgene plasmid #14622) from Roger Davis (University of Massachusetts Medical School, Worcester, USA), ZsProSensor-1 proteasome sensor (Clontech, Mountain View, CA), VSVG tagged with GFP (Jennifer Lippincott-Schwartz, NICHD, NIH, Bethesda, USA), Cdc42 (A. Hall, Sloan-Kettering Institute, New York, NY, USA), P-glycoprotein wild type, G268V and DY490 mutants (David M. Clarke, University of Toronto, Canada) and hERG wild type and G601S mutant (Alvin Shrier, McGill University, Montreal, Canada). The reagents used include: VX-809 (Selleckchem, Germany), JNKi II (SP600125), JNKi IX and JNKi XI (Merck Millipore, Germany), oxozeaenol (Tocris Bioscience, UK), siRNAs (see *Supplementary file 6*), lipofectamine 2000 (Thermo Fisher Scientific) and ECL (Luminata crescendo from Merck Millipore).

## Transcriptional basis of corrector MOA

To understand if the proteostatic correctors have a transcriptional component in their corrector MOA, the sensitivity of their corrector action to actinomycin D, an inhibitor of transcription, was tested. To this end, we first analysed the kinetics of action of the corrector drugs in modulating F508del-CFTR proteostasis. HeLa cells stably expressing HA-tagged F508del-CFTR (*Okiyoneda et al., 2010*) were treated with selected corrector drugs (from MANTRA dataset) that resulted in a detectable increase in the intensity of band B already after 3 hr, while the effects on band C levels were detectable only after 12–24 hr of drug treatment (not shown). In order to minimize the toxic effects of actinomycin D on cell physiology and since the increase in band C was always preceded by an increase in band B, we decided to monitor the actinomycin D sensitivity of the corrector drugs towards the regulation of F508del-CFTR proteostasis by monitoring the early changes in band B levels. To this end, HeLa cells stably expressing HA-tagged F508del-CFTR were treated either with corrector drugs alone ([Chlorzoxazone [50 μM], Glafenine [50 μM], Trichostatin-A [500 nM], Dexamethasone [500 nM], Doxorubicin [500 nM]) or along with 10 ug/ml actinomycin-D for 3 hr and the levels of band B were determined by western blotting. While actinomycin D treatment did not have any effect on the increase in band B levels resulting from treatment with VX-325 or Corr-4a, it significantly reduced band B levels in other cases (not shown). This suggests that the corrector drugs (except VX-325 and Corr-4a) are proteostatic regulators that act by inducing transcriptional changes in the cell.

## Analysis of corrector-induced gene expression changes by microarray

Polarised CFBE41o-cells cultured at the air–liquid interface were treated with the corrector drugs of interest (CFBE dataset, *Table 1*) for 24 hr. Total RNA was extracted and hybridization was carried out on to Whole Human Genome 44 K arrays (Agilent Technologies, product G4112A) following the manufacturer's protocol. See (*Zhang et al., 2012*) for experimental details. The microarray data for ouabain and low temperature treatments have been published elsewhere (*Zhang et al., 2012*).

## FIT analysis of microarray profiles

The microarrays from the connectivity map database (https://www.broadinstitute.org/cmap/) were processed to produce prototype ranked lists (PRLs) (*Iorio et al., 2010*). In these PRLs, cell line-specific responses are diluted, thus summarising consensual transcriptional responses to drug treatment. In each PRL, microarray probe-sets are ordered from the most upregulated to most downregulated one. We downloaded PRLs for the whole panel of small molecules in the connectivity map (www.connectivitymap.org) from which the MANTRA database is derived (http://mantra.tigem.it/). We used these in conjunction with ranked lists of probe sets based on fold changes (and assembled following the guidelines provided in [*Iorio et al., 2010*]) from microarray profiles that we generated in house (CFBE dataset).

The FIT analysis identifies microarray probe-sets that tend to respond consistently to a group of drugs (see also [*Iorio et al., 2010*] for description of a similar method). The top and bottom 20% of the probe-sets (corresponding to the up- and downregulated probe-sets, respectively) were used for the analysis. The 20% cut-off was used since the merging of individual gene expression profiles into PRLs precludes the application of other thresholds based on fold change (or p-value) to identify

significantly differentially expressed genes. To build a null model against which the significance of the final genes sets can be tested (as detailed below), a fixed number of PRLs (N) from the MANTRA dataset were randomly selected and the upregulated or downregulated probe-sets from this selection were intersected by varying the fuzzy cut-off threshold (i.e. the ratio of drugs that a given probe-set should transcriptionally respond to, in order to be considered 'consistently' regulated, hence to be included in the fuzzy intersection). After 1000 of these iterations, we derived an empirical null distribution of the number of probes included in the resulting fuzzy intersections and used it for p-value assignments (*Figure 1B*). For the CFBE dataset (generated on an Agilent platform, which is different from that used for the connectivity map and MANTRA database), we derived this null distribution by randomly permuting all the individual probes. Finally, we determined the optimal fuzzy cut-off values for the transcriptional profiles elicited by the corrector drugs (11 contained in MANTRA and 13 in the CFBE dataset). Briefly, we selected the value such that the number of probes present in the final fuzzy intersection was at least threefold higher than that expected by random chance and its p-value < 0.05 (according to the computed null models). By using this method, no significantly upregulated probes from the MANTRA dataset were identified across all of the range of tested fuzzy cut-offs. For the downregulated probe-sets a fuzzy cut-off of 8 (out of 11 corrector drugs) or above produced significant fuzzy intersection of probe-sets. For the CFBE dataset, a significant cut-off of 6 drugs (out of 13) and above was identified. To optimise the selection of these cut-offs further, we chose the maximal cut-off yielding a fuzzy intersection of probe-sets enriched in one or more Gene Ontology terms. With this criterion, we obtained a final cut-off value of 8 for the MANTRA downregulated probe set and cut-off of 9 for the CFBE dataset. Intersecting the corrector-induced gene expression profiles using this optimal fuzzy cut-off resulted in 541 upregulated probe-sets (mapping 402 unique genes) and 191 downregulated probe-sets (mapping to 117 unique genes) for the CFBE dataset, and 108 downregulated probe-sets (mapping 102 genes) for the MANTRA dataset. Note that most of the CORE genes (519 out of the 621 CORE genes) are derived from the CFBE dataset. This, we suppose, is due to the use of PRLs in the case of cMAP dataset and use of data derived from a single cell line in the case of CFBE dataset. The use of single cell line-derived data can potentially lead to high number of false positives since perturbation-independent response of cell lines to treatments is usually stronger than the perturbation-dependent response (*Iorio et al., 2010*).

We finally validated the optimal number of drugs that need to be considered for a fuzzy cut-off of 70% (corresponding to 8 out 11 drugs cut-off from the MANTRA dataset), providing a minimum number of false positives in the intersection (i.e. genes expected to be contained in the resulting intersections by random chance). This was performed by a permutation test where, in a series of iterations, the fuzzy cut-off is kept constant and the number of randomly selected drugs varied within a given range (specifically from 1 to 20). At each of these iterations, we computed the cardinality of the resulting fuzzy intersections, observing that this value reached a plateau at 10 drugs (*Figure 1C*), which suggests that the number of drugs that was used in the analysis (i.e. 11 drugs in the cMAP dataset) was fairly close to the optimal level.

## Protein-protein interaction

The protein-protein interactions were downloaded from the STRING database (http://string-db.org/) (*Franceschini et al., 2013*), and those with a confidence level of >0.7 were used for the analysis. To build the proteostasis gene (PG) dataset, we included known proteostatic regulators of CFTR, that is, proteins where their expression/activity level changes have been shown to affect CFTR proteostasis. We also included the interactors of CFTR and CF pathology related genes/proteins present in GeneGO Metaminer Cystic Fibrosis database (see *Supplementary file 4* for the list of the proteostasis genes). The number of interactions observed among the CORE gene dataset and the proteostasis gene dataset as well as among the CORE gene dataset were more than expected on a random basis and were statistically significant. For details on the statistical test used see (*Franceschini et al., 2013*).

## Ingenuity pathway analysis

The gene sets were analyzed using the CORE analysis application of the Ingenuity pathway analysis, a web-based software application. The default settings of the analysis were used. Each network had

an assigned significance score based on the p-value (calculated using Fischer's exact test) for the probability of finding the focus genes in a set of genes randomly selected from the global molecular network. The upregulated and downregulated genes of the CFBE dataset and the downregulated genes of the cMAP dataset were analyzed separately and also together, to infer common pathways or networks embedded among them.

## Cell lysis, western blotting and analysis

Cells were washed three times in ice-cold Dulbecco's phosphate-buffered saline, and lysed in RIPA buffer (150 mM NaCl, 1% Triton X-100, 0.5% deoxycholic acid, 0.1% SDS, 20 mM Tris-HCl, pH 7.4), supplemented with protease inhibitor cocktail and phosphatase inhibitors. The lysates were clarified by centrifugation at 15,000 x g for 15 min, and the supernatants were resolved by SDS-PAGE. BCA Protein Assay kit (Pierce) was used to quantitate protein levels before loading. The western blots were developed with appropriate antibodies and using ECL. The blots were then exposed to x-ray films and exposure time was varied to obtain optimal signal. The X-ray films were then scanned and the bands were quantitated using ImageJ gel-analysis tool (see *Figure 2—figure supplement 2A–D*). The protein concentration and the exposures used for quantitation of the blots were optimized to be in a linear range of detection (*Figure 2—figure supplement 1E,F*).

## Biochemical screening assay

Each gene was targeted by three siRNAs and as control non-targeting siRNAs provided by the manufacturer were used (see *Supplementary file 6* for list of siRNAs used). A gene was considered as active if: (1) at least two different siRNAs targeting a gene gave concordant changes in the levels of band C that was >2 SD from the mean value of the control siRNAs and (2) the change in band C levels was ± 20% of the level of band C obtained with the control siRNAs. Those genes that increased band C levels significantly upon their downregulation were termed anti-correction genes and those that decreased band C levels were termed pro-correction genes.

A potential problem in siRNA-based experiments is the possibility of off-target effects. The specificity of the observed effects on F508del-CFTR proteostasis are supported by the following lines of evidence: (1) The quantitation of the on-target effect of the siRNAs by RT-PCR (see *Figure 2—figure supplement 3*), which shows that treatment with siRNAs brings down the transcript levels of the target genes to ~10–30% of the levels present in cells treated with non-targeting siRNAs; (2) The use of at least three different siRNAs (individually) for the screenings, the majority of which showed concordant results on proteostasis. Moreover, for selected genes (MLK3, CAMKK2, RNF215, NUP50 and CD2BP2), the findings from the screening studies were reinforced using additional siRNAs (for instance, five additional siRNAs for MLK3; see *Supplementary file 6* for details of the siRNAs used) that showed similar effects on proteostasis of F508del-CFTR (not shown). (3) The coherent effects of the siRNA targeting different genes of signaling pathways supports the functional significance of the effects of each siRNA (see *Figure 3A,B* and *Figure 3—figure supplement 1B*). (4) Overexpression of identified hits show an opposite effect on proteostasis to that of the siRNAs (see below *Figure 3C*). (5) Finally, there was a positive correlation between the concentration of siRNAs used and the effect on proteostasis (*Figure 2—figure supplement 3*). All these evidence together confirm the specificity of the observed effect of the siRNAs on proteostasis.

## Immunoprecipitation

HeLa cells cultured in 10-cm plates (80% confluence) were treated with appropriate corrector drugs for 24 hr. The cells then were washed three times in ice-cold Dulbecco's phosphate-buffered saline, and lysed in immunoprecipitation buffer (150 mM NaCl, 1% Triton X-100, 20 mM Tris-HCl, pH 7.4) on ice for 30 min. The lysates were clarified by centrifugation at 15,000 x g for 15 min, and the protein content of the supernatants quantitated by BCA Protein Assay kit (Pierce). Equal amounts of proteins from control and treated cell lysates were incubated with Protein-G sepharose beads conjugated with anti-HA antibody (Sigma) overnight at 4°C. The beads were then washed in the immunoprecipitation buffer five times and the bound proteins eluted with HA-peptide (Sigma) at a concentration of 100 µg/ml. The eluted proteins were then resolved by SDS-PAGE and immunoblotted.

## Partial trypsin digestion of CFTR

The trypsin digestion assay was similar to that described previously (*Zhang et al., 1998*). Cells were grown in a 10-cm plate and post-treatment were washed three times with 10 mL phosphate-buffered saline (PBS). They were then scraped in 5 mL PBS, and pelleted at 500 x g for 5 min in 4°C. The cell pellet was resuspended in 1 mL of hypertonic buffer (250 mM sucrose, 10 mM Hepes, pH 7.2) and the cells were then homogenized using a ball bearing homogenizer. The nuclei and unbroken cells were removed by centrifugation at 600 x g for 15 min. The membranes were then pelleted by centrifugation at 100,000 x g for 30 min, and then resuspended in digestion buffer (40 mM Tris pH 7.4, 2 mM MgCl$_2$, 0.1 mM EDTA). Then membranes corresponding to 50 µg of protein were incubated with different concentrations of trypsin (1 to 50 µg/ml) on ice for 15 min. The reactions were stopped with the addition of soya bean trypsin inhibitor (Sigma) to a final concentration of 1 mM, and the samples were immediately denatured in sample buffer (62.5 mM Tris-HCL, pH 6.8, 2% SDS, 10% glycerol, 0.001% bromophenol, 125 mM dithiothreitol) at 37°C for 30 min. The samples were resolved on 4-16% gradient SDS-PAGE (Tris-glycine) and transferred onto nitrocellulose membranes. These membranes were developed with the 3G11 anti-CFTR antibodies (that recognize nucleotide-binding domain 1 - NBD1) or the M3A7 clone (that recognizes nucleotide-binding domain 2 -NBD2).

## Plasma membrane quality control (PQC) assay

The PQC assay was essentially as described previously (*Okiyoneda et al., 2010*). CFBE cells were untreated or treated with siRNAs for 72 hr and for the final 31 hr they were kept at low temperature (26°C) and for an additional 5 hr at 26°C with CHX (100 µg/ml). Then, the cells were shifted to 37°C for 1.5 hr with 100 µg/ml CHX before the turnover measurements started at 37°C. The cells were lysed at 0, 1, 3 and 5 hr and the kinetics of degradation of band C was examined by immunoblotting.

## Halide sensitive YFP assay for CFTR activity

Twenty-four hours after plating, the CFBE cells that stably expressed halide sensitive YFP were incubated with the test compounds at 37°C for 48 hr. At the time of the assay, the cells were washed with PBS (containing 137 mM NaCl, 2.7 mM KCl, 8.1 mM Na$_2$HPO$_4$, 1.5 mM KH$_2$PO$_4$, 1 mM CaCl$_2$, 0.5 mM MgCl$_2$) and stimulated for 30 min with 20 µm forskolin and 50 µm genistein. The cells were then transferred to a Zeiss LSM700 confocal microscope, where the images were acquired with a 20x objective (0.50 NA) and with an open pinhole (459 µm) at a rate of 330 ms/frame (each frame corresponding to 159.42 µm x 159.42 µm), at ambient temperature. The excitation laser line 488 nm was used at 2% efficiency coupled to a dual beam splitter (621 nm) for detection. The images (8-bit) were acquired in a 512x512 format with no averaging to maximize the speed of acquisition. Each assay consisted of a continuous 300-s fluorescence reading with 30 s before and the rest after injection of an iodide-containing solution (PBS with Cl$^-$ replaced by I$^-$; final I$^-$ concentration in the well 100 mM). To determine the fluorescence-quenching rate associated with I$^-$ influx, the final 200 s of the data for each well were fitted with a mono-exponential decay, and the decay constant K was calculated using GraphPad Prism software.

## Ussing chamber assay for short circuit current recordings

Short-circuit current ($I_{sc}$) was measured across monolayers in modified Ussing chambers. CFBE41o$^-$ cells (1x10$^6$) were seededonto 12-mm fibronectin-coated Snapwell inserts (Corning) and the apical medium was removed after 24 hr to establish an air-liquid interface. Transepithelial resistance was monitored using an EVOM epithelial volt-ohm meter and cells were used when the transepithelial resistance was 300–400 Ω.cm$^2$. CFBE41o$^-$ monolayers were treated on both sides with optiMEM medium containing 2% (v/v) FBS and one of the following compounds: 0.1% DMSO (negative control), or compounds at the stated dosage for 48 hr before being mounted in EasyMount chambers and voltage-clamped using a VCCMC6 multichannel current-voltage clamp (Physiologic Instruments). The apical membrane conductance was functionally isolated by permeabilising the basolateral membrane with 200 µg/ml nystatin and imposing an apical-to-basolateral Cl$^-$ gradient. The basolateral bathing solution contained 1.2 mM NaCl, 115 mM Na-gluconate, 25 mM NaHCO$_3$, 1.2 mM MgCl$_2$,4 mM CaCl$_2$, 2.4 mM KH$_2$PO$_4$, 1.24 mM K$_2$HPO$_4$ and 10 mM glucose (pH 7.4). The CaCl$_2$concentration was increased to 4mM to compensate for the chelation of calcium by gluconate. The apical bathing

solution contained 115 mM NaCl, 25 mM NaHCO$_3$, 1.2 mM MgCl$_2$, 1.2 mM CaCl$_2$, 2.4 mM KH$_2$PO$_4$, 1.24 mM K$_2$HPO$_4$ and 10 mM mannitol (pH 7.4). The apical solution contained mannitol instead of glucose to eliminate currents mediated by Na-glucose co-transport. Successful permeabilization of the basolateral membrane was obvious from the reversal of $I_{sc}$ under these conditions. Solutions were continuously gassed and stirred with 95% O$_2$-5% CO$_2$ and maintained at 37°C. Ag/AgCl reference electrodes were used to measure transepithelial voltage and pass current. Pulses (1 mV amplitude, 1 s duration) were delivered every 90 s to monitor resistance. The voltage clamps were connectedto a PowerLab/8SP interface for data collection. CFTR was activated by adding 10 µM forskolin to the apical bathing solution.

## Acknowledgements

We acknowledge the financial support of Italian Cystic Fibrosis Research Foundation (FFC#2 2014), Fondazione Telethon, AIRC (Italian Association for Cancer Research, IG 10593), the MIUR Project 'FaReBio di Qualità', the PON projects no. 01/00117 and 01-00862, PONa3-00025 (BIOforIU), PNR-CNR Aging Program 2012-2014 and Progetto Bandiera 'Epigen' to AL, Fondazione Telethon and Ministero della Salute, Italy to DdB. FCi was supported by a postdoctoral fellowship from Pasteur Institute Cenci-Bolognetti Foundation, Biology and Biotechnology Department 'Charles Darwin' of Sapienza University of Rome, and partially funded by the Telethon Institute of Genetics and Medicine. GC was supported by a fellowship from Cystic Fibrosis Canada and by the Canadian Institutes for Health Research. DYT is the Canada Research Chair in Molecular Genetics. DYT acknowledges the support from Canada Foundation for Innovation and the Canadian Institutes for Health Research (MOP-89779). We thank the Bioinformatic core Facility of the Telethon Institute of Genetics and Medicine, for help with bioinformatics analysis and the Bioimaging Facility of the Institute of Protein Biochemistry for help with the image acquisition.

## Additional information

### Funding

| Funder | Grant reference number | Author |
| --- | --- | --- |
| Fondazine per la ricerca sulla fibrosi cistica-onlus | FFC#2 2014 | Seetharaman Parashuraman<br>Alberto Luini |
| Canadian Institutes of Health Research | | Fabiana Ciciriello<br>David Y Thomas |
| Cystic Fibrosis Canada | | Graeme Carlile |
| Canada Foundation for Innovation | MOP-89779 | David Y Thomas |
| Fondazione Telethon | | Diego Di Bernardo<br>Alberto Luini |
| Ministero della Salute | | Diego Di Bernardo |
| Ministero dell'Istruzione, dell'Università e della Ricerca | 01/00117 and 01-00862, PONa3-00025, A COmputational approach for the identification of the Secondary Mechanism of action of drugs: application to Cystic Fibrosis (COSM) | Alberto Luini |
| Associazione Italiana per la Ricerca sul Cancro | IG 10593 and 15767 | Alberto Luini |

The funders had no role in study design, data collection and interpretation, or the decision to submit the work for publication.

### Author contributions

RNH, SP, Designed and developed the idea, conducted the experiments largely at IBP, CNR and wrote the manuscript, Conception and design, Acquisition of data, Analysis and interpretation of

data, Drafting or revising the article; FI, Contributed to conception of the computational method, the bioinformatic analysis and the writing of the manuscript; FCi, Produced the microarray data and involved in microarray data analysis; FCa, VB, MP, Involved in bioinformatic-analysis; AC, DC, Involved in microarray data analysis; AS, LB, Conducted the experiments; GC, LG, Helped in the anion conductance assays; DYT, Produced the microarray data and helped in the anion conductance assays; DDB, Involved in microarray data analysis bioinformatic-analysis; AL, Designed, developed the idea and wrote the manuscript, Conception and design, Analysis and interpretation of data, Drafting or revising the article

### Author ORCIDs

Ramanath Narayana Hegde, http://orcid.org/0000-0001-7224-3544

## Additional files

### Supplementary files

• Supplementary file 1. The microarray data for the CFBE dataset (related to *Figure 1*).

• Supplementary file 2. Drug communities containing the 11 MANTRA drugs extracted from the MANTRA drug similarity network assembled in (Iorio et al., 2010) (related to *Figure 1—figure supplement 1*).

• Supplementary file 3. The CORE genes obtained using the FIT analysis and list of genes selected for experimental validation (related to *Figures 1* and *2*).

• Supplementary file 4. The proteostasis genes used for this study (related to *Figure 1*).

• Supplementary file 5. The regulatory and proteostasis genes present in the CORE genes list (related to *Figure 1*).

• Supplementary file 6. The siRNAs used in this study (related to *Figure 2*).

### Major datasets

The following previously published dataset was used:

| Author(s) | Year | Dataset title | Dataset URL | Database, license, and accessibility information |
|---|---|---|---|---|
| Iorio F, Bosotti R, Scacheri E, Belcastro V, Mithbaokar P, Ferriero R, Murino L, Tagliaferri R, Brunetti-Pierri N, Isacchi A, di Bernardo D | 2010 | Discovery of drug mode of action and drug repositioning from transcriptional responses | http://mantra.tigem.it/ | Available at www. mantra.tigem.it on request |

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
