## [Decision Letter]

Thank you for submitting your work entitled "Unravelling druggable signalling networks that control F508del-CFTR proteostasis" for peer review at *eLife*. Your submission has been favorably evaluated by James Kadonaga (Senior editor) and three reviewers, one of whom, Suzanne Pfeffer, is a member of our Board of Reviewing Editors.

The reviewers have discussed the reviews with one another and the Reviewing editor has drafted this decision to help you prepare a revised submission.

There is great interest in identifying pharmaco-chaperones and other molecules to help patients better fold mutant membrane glycoproteins that when misfolded, lead to human disease. Here the authors focus on the major mutant form of CFTR and combine the data obtained from multiple microarray analyses in the presence of a number of proteostasis regulators (that act by independent mechanisms) to identify overlapping pathways that may not otherwise be revealed. They then use siRNAs and/or chemical blockers and find a set that enhance protein levels, protein export to the Golgi, and when combined with pharmaco-chaperones yield even higher levels of exported CFTR protein. The paper is well written, the findings are of broad interest and with minor textual modifications the work should be presented in *eLife*.

1) Figure 2 legend and text: siRNAs in Figure 2 increase band C (are good). The related text "downregulating negative correction" (2A) can be confusing. Please find a better way to explain this that will be easier for the reader throughout the text. In E, these are called "positive correction" hits.

2) In the subsection “ROS and the CF modifiers TNF-α, TGF-β enhance F508del-CFTR degradation in an MLK3–dependent fashion”, "within a few minutes" seemed shocking for the change observed; H_2_O_2_ treatment was apparently for 30 min (Figure 5) which is much more than a few. Please correct text.

3) Figure 6: Please label x-axis with numbers for concentrations.

4) Figure 6—figure supplement 1 shows JNK inhibitors for other mutant proteins, some helped some hurt. It would be helpful if the authors could include some non-mutant, endogenous non-mutant membrane proteins in their pathway analysis to see if their levels or turnover change under one or two of the corrective conditions reported.

5) The starting point of the study is a bioinformatics method ("FIT") developed by the authors. Please clarify the motivation for the development of this approach and compare with existing approaches. One conceptual problem is the assumption that all compounds should act through the same mechanism and therefore share transcriptional effects. What if there are 2 distinct mechanisms without shared transcripts? Then such transcripts would not be detected by the authors' approach, but could be detected by hierarchical clustering already implemented in freely available software (e.g. GENE-E). If the compounds are proteostasis regulators and act transcriptionally as the authors claim, one might have expected the "CORE" genes detected by the method to show statistically significant enrichment for proteostasis-related gene sets (by GO-term or GSEA analysis). Please discuss.

6) Most of the characterization of compound and siRNA effects relies on a single assay: the quantification of western blot bands B and C. It is not clear how reliable this assay is. E.g. in Figure 2 first lane, no "C" band is visible, however the authors quantified it, raising questions regarding background correction. In Figure 6, bands B and C are part of a continuous smear, raising concerns about accurate quantification. In Figure 6, error bars are very large, raising concerns regarding reproducibility. In Figure 6—figure supplement 2, functional readouts for CFTR activity are used, but they do not seem to correlate well with the "Band C" assay, and in fact no single compound or siRNA seems to significantly outperform VX-809 in the functional assay (whereas the "band C" assay suggested otherwise). This highlights that any major conclusion in the paper should be moderated – please discuss this issue.

7) The authors extensively use siRNA to investigate the effect of knocking down specific transcripts on expression of mature CFTR. siRNAs are notorious for off-target effects, and cannot be assumed to active unless this is directly shown. This problem is illustrated in Figure 3, where the two negative control siRNAs already show drastic differences in band B, although both are presumably not affecting any transcript. For any major conclusions in the paper, on-target effect needs to be quantified for the siRNAs used, and off-target effects need to be excluded by either showing dose-dependent effect of a series of siRNAs with increasing knockdown efficiency, or rescue of the wild-type phenotype by expressing a knockdown-resistant version of the transcript.

8) In the Introduction, the authors state that "systematic exploration of the signalling pathways that regulate the initial stages of the proteostasis viz. the folding and degradation of proteins is lacking." This is ignoring a large body of work into the cellular signalling pathways associated with cytosolic proteostasis (heat shock response) and the ER (unfolded protein response, UPR). The therapeutic potential of targeting these pathways is actively being explored by other groups (e.g. see PMID: 23647985 for a review). Please add.

9) Also in the Introduction, the authors state that "Unfortunately, the effects of the available correctors are too weak to be of clinical interest". However, Lumacaftor (VX-809) in combination with ivacaftor is FDA approved for F508delta CF patients, based on efficacy in clinical trials (improved lung function over placebo): http://www.fda.gov/NewsEvents/Newsroom/PressAnnouncements/ucm453565.htm

10) The authors conclude that increase in CFTR levels cannot be due to increased synthesis based on the unchanged mRNA levels. They cannot exclude effects on translational efficiency.

11) Figure 2: triangle for mitoxantrone concentration appears flipped? Actual concentrations should be indicated in the figure or legend.

12) Official gene symbols are not consistently used, and labels for siRNAs in figures do not always correspond to the labels in [Supplementary-material SD5-data], where the siRNA sequences are listed.

13) The trypsin susceptibility assay may capture some, but not all differences in folding that are relevant for ERAD/QC. In general, the evidence the authors provide to characterize the mechanism and nature of MLK3 regulation should be presented with all the relevant caveats.

14) The finding that siRNA knockdown of TAK1 does not equal the effect of oxozeaenol is not by itself proof that oxozeaenol does not act through TAK1, since pharmacological inhibition and siRNA knockdown of kinases can have distinct consequences.

15) It would be nice but is not essential to show this efficacy in primary lung tissue from patients. It is well established that the influence of the proteostasis regulators on F508del-CFTR folding and trafficking is not always repeatable when translating the work from easily culturable cells to lung cells derived from patients-the standard now used by those commercial organizations developing CF drugs.

16) It also seems important to show (or at least discuss) that the 20-fold increases in the C-band has a functional significance in the presence of a channel opening pharmacologic agent. The F508del-CFTR is deficient as a chloride channel even when it is properly folded and trafficked to the plasma membrane of polarized epithelial cells.

---

## [Author Response]

1) Figure 2 legend and text: siRNAs in Figure 2 increase band C (are good). The related text "downregulating negative correction" (2A) can be confusing. Please find a better way to explain this that will be easier for the reader throughout the text. In E, these are called "positive correction" hits.

We thank the reviewers for pointing out that some terminology needed to be clarified. In order to avoid confusion, we have changed “positive correction” and “negative correction genes” to “pro-correction” and “anti-correction genes” respectively. Those genes that, upon downregulation, increase band C levels are now called “anti-correction genes” and those that decrease band C levels upon downregulation are called “pro-correction genes”.

The sentence in the legend to Figure 2 now reads as follows:

“The fold change in the levels of band C obtained by downregulating anti-correction (A) and pro-correction (D) genes and the fold change in levels of band B (B) and band C/ band B ratio (C) after downregulation of the anti-correction genes are shown.”

*2) In the subsection “ROS and the CF modifiers TNF-α, TGF-β enhance F508del-CFTR degradation in an MLK3–dependent fashion”, "within a few minutes" seemed shocking for the change observed; H2O2 treatment was apparently for 30 min (Figure 5) which is much more than a few. Please correct text.*

We have now modified the text to read as follows:

“The effects of H_2_O_2_ at non-toxic concentrations were dramatic, with a marked drop of the F508del-CFTR levels within 30 min (Figure 5).”

*3) Figure 6: Please label x-axis with numbers for concentrations.*

We were unable to mark the concentrations with numbers on the x-axis in panel B–D, since the effective concentrations of the inhibitors varied considerably (as mentioned in panel A). For instance, while the effective concentration of JNKi II ranged from 6.25 to 5 μM, that of another inhibitor – JNKi IX – ranged from 1.25 to 10 μM. Therefore, in order to make it easier to follow the figures, we present the concentrations as values normalized to the maximal effective concentration of the drug. This is now mentioned in the figure legends along with a note referring to panel A where the actual concentrations used are indicated.

4) Figure 6—figure supplement 1 shows JNK inhibitors for other mutant proteins, some helped some hurt. It would be helpful if the authors could include some non-mutant, endogenous non-mutant membrane proteins in their pathway analysis to see if their levels or turnover change under one or two of the corrective conditions reported.

The loading control that we had used (Na+-K+ ATPase) already indicates that the pathway does not regulate the proteostasis of all endogenous non-mutant proteins. We have now also included data on other non-mutant endogenous proteins like E-cadherin, IGF1Rß and EGFR (Figure 6—figure supplement 1). The results show that proteostasis is not regulated by the MLK3-JNK pathway.

*5) The starting point of the study is a bioinformatics method ("FIT") developed by the authors. Please clarify the motivation for the development of this approach and compare with existing approaches. One conceptual problem is the assumption that all compounds should act through the same mechanism and therefore share transcriptional effects. What if there are 2 distinct mechanisms without shared transcripts? Then such transcripts would not be detected by the authors' approach, but could be detected by hierarchical clustering already implemented in freely available software (e.g. GENE-E).*

We had initially analysed the corrector drugs by classical, as well as state-of-the-art clustering methods, without obtaining meaningful results though. In our attempts to apply clustering methods, we had analyzed the MANTRA and CFBE datasets separately, for technical reasons. We now provide the results of these analyses in Figure 1—figure supplement 1and [Supplementary-material SD2-data].

The MANTRA dataset: This set of differential expression profiles was downloaded from the Connectivity Map (cMap) database and was pre-processed as described before (Iorio et al., PNAS 2010). Due to the heterogeneous experimental settings through which the data present in cMap were generated (i.e. multiple batches, different dosages, hyper/hypo-replicated experiments with possibly over/poorly-represented cell-line-specific responses), a rank-merging procedure able to dilute batch-, dose- and cell-line-specific effects on the transcriptional responses of a drug was applied (as described in Iorio et al., PNAS 2010). This results in a simple genome-wide ranked-list of genes for each drug, which summarizes the consensual transcriptional responses to the drug across multiple settings.

While this method improves the comparability of the drugs (Iorio et al., PNAS 2010), it also results in loss of information about fold changes in transcript levels and p-values. Thus, the resulting lists are not suitable for comparisons and cluster analyses made with traditional metrics (such as Correlation or Euclidian distance) and classic partitional or agglomerative/hierarchical clustering methods (Iorio et al., PNAS 2010). So we have not applied a traditional hierarchical cluster analysis (or GENE-E as mentioned by the reviewer). However, we attempted to overcome this issue by an alternative way to “cluster” the profiles. In the past, in an effort to cluster the cMap drugs based on their transcriptional response similarity, we had designed a novel Gene set enrichment analysis (GSEA) based metric to obtain similarity scores between ranked lists (see Iorio et al., PNAS 2010 for details). This study resulted in a network that contains clusters of densely interconnected drugs (named drug communities), whose corresponding ranked-lists had high “similarity” scores. Since this network is basically the output of a state-of-the-art method to cluster the ranked-lists of genes derived from cMap, we have mined this network in an attempt to cluster the F508del-CFTR corrector drugs. However, results from this analysis (included in the newly assembled Figure 1—figure supplement 1and [Supplementary-material SD2-data]) show that: (i) most of the F508del-CFTR corrector drugs clustered into separate drug communities enriched for their principal MOA and (ii) they did not reveal meaningful commonalities that might be related to their effects as correctors.

The CFBE dataset: This dataset could be analyzed by traditional clustering methods since the expression profiles were obtained from homogeneous experimental settings (and therefore it was not necessary to have it processed as the MANTRA dataset). Nevertheless, again, we did not get any meaningful or statistically significant clusters.

These results led us to reason that the drugs have primary MOAs that are heterogeneous and are most likely unrelated to correction. These primary MOAs have a stronger influence on the transcriptional signature than the correction-relevant secondary MOA. Thus, the impossibility of obtaining significant clusters is due to the heterogeneous primary MOAs precluding meaningful clustering. This led us to develop the FIT method in order to detect the common transcriptional signature associated with their secondary MOAs.

The aim of the FIT method is to harvest a set of genes that is consistently differentially expressed by majority of the F508del-CFTR correctors. The presence of common genes shared by most of the drugs (more than 70% of the correctors) as identified by FIT method indicates that, indeed, there is an underlying commonality amongst the effects of the correctors, and further analysisshows that there are several common secondary MOAs that are shared by most of the correctors. Notably, the fact that these MOAs are shared by several correctors suggests that they might be among the most functionally relevant. As the reviewers point out, nevertheless, it is also possible that there could be effective idiosyncratic mechanisms by which individual drugs or small groups of drugs may affect F508del-CFTR proteostasis that the FIT method might we have missed.

We have now added a note in the Results section referring to these analyses:

“To extract the correction-related transcriptional effects from those due to primary effects of the correctors […] these drugs obscures the potential clustering of drug signatures based on their secondary correction-relevant MOAs.”

*If the compounds are proteostasis regulators and act transcriptionally as the authors claim, one might have expected the "CORE" genes detected by the method to show statistically significant enrichment for proteostasis-related gene sets (by GO-term or GSEA analysis). Please discuss.*

Proteostasis related genes were included in the CORE genes resulting from the FIT method (see [Supplementary-material SD5-data]). However, they were not significantly enriched, indicating that changes in the expression levels of a large fraction of the proteostasis machinery genes is not part of the MOA of the proteostasis regulator drugs. It remains possible, however, that a few crucial proteostasis genes might be regulated by these correctors, and that the number of these genes might too small to result in a statistically significant enrichment of this gene group.

Additionally, it is also possible that the regulatory effects on proteostasis induced by the corrector drugs are mediated by changes in expression of kinases or other regulatory proteins, that exert their effects by modulating the activity of the proteostasis machinery components, rather than their levels. Our study of the interaction of HOP with F508del-CFTR under conditions of MLK3 pathway inhibition (Figure 4—figure supplement 1) suggests that, at least for HOP, this is indeed the case, namely, that this pathway can influence the interaction of HOP with F508del-CFTR (probably via phosphorylation) without directly affecting the levels of the HOP protein.

Also of note is that there are significant interactions between the CORE genes and the known proteostasis gene pools (Figure 1), suggesting that the CORE genes might regulate the proteostasis machinery *also* through such interactions. The nature of these interactions remains an issue for further studies.

We have now added a note in the Results section mentioning the same:

“A search proteostasis components among CORE genes retrieved 48 folding/ degradation and 24 transport-machinery components ([Supplementary-material SD5-data]), some of which are known to be involved in F508del-CFTR proteostasis. […] or that the corrector drugs act by modulating the expression of regulatory genes/pathways that act post-translationally on the proteostasis machinery (see results from the screening below). “

*6) Most of the characterization of compound and siRNA effects relies on a single assay: the quantification of western blot bands B and C. It is not clear how reliable this assay is. E.g. in Figure 2 first lane, no "C" band is visible, however the authors quantified it, raising questions regarding background correction. In Figure 6, bands B and C are part of a continuous smear, raising concerns about accurate quantification. In Figure 6, error bars are very large, raising concerns regarding reproducibility.*

The seeming undetectability of band C signal in Figure 2 is due to technical reasons, viz. the large difference between the intensities of band C in control and treated cells. In order to get all the bands in a detectable as well as linear range for quantitation several exposures of the blots were used and quantitations were restricted to those exposures where the intensities of the bands under study were in the linear range. In the experiments in Figure 2, the quantitations were done using blots where the time of exposure was adjusted so as to visualize band C in the linear range of measurement under all treatment conditions (see Figure 2—figure supplement 1 for details of quantitation). The blots presented in the figures were the ones that were underexposed so that the changes in the band C levels can be easily appreciated visually. In this particular case (Figure 2) we have now added an additional blot with higher exposure (that was included in the quantitation) where band C can be clearly seen in the control condition.

Regarding the presence of a smear of band C in several samples (Figure 6), we do agree that this might actually lead to an underestimation of the correction levels. Even though the region of interest (ROI) used for quantitation was relatively broad to include the majority of the area of the smear in the vicinity of band C (see Figure 2—figure supplement 1 for details of quantitation), in some cases a portion of the smear was outside the ROI and thus resulting in the underestimation of the correction levels. Thus the levels presented here are conservative estimates.

For the variability of some of the results (see Figure 6), we note that most error bars are acceptable (close to 10% of the corresponding mean value), and are larger at the highest effective concentration of some of the drugs used. We have no clear explanation for this, apart from the possibility that the technical problems discussed above (e.g., the large smear of band C) might be accentuated at high levels of correction.

*In Figure 6—figure supplement 2, functional readouts for CFTR activity are used, but they do not seem to correlate well with the "Band C" assay, and in fact no single compound or siRNA seems to significantly outperform VX-809 in the functional assay (whereas the "band C" assay suggested otherwise). This highlights that any major conclusion in the paper should be moderated – please discuss this issue.*

As discussed in the text, the goal of this study is to *selectively* elucidate the signalling networks that control the proteostasis (i.e., degradation, folding and export) machinery acting on F508del-CFTR, rather than controlling the overall function of F508del-CFTR, which of course depends also on the channel properties of this mutant. We think that proteostasis and channel gating defects of F508del-CFTR are different aspects of CF that are regulated differently and need to be tackled separately for clarity of analysis.

Based on the same concept, this study does not aim to generate efficient correctors ready for clinical use. This goal might not be realistic considering our still limited understanding of the biological mechanisms underlying the F508del-CFTR loss of function. Rather, it aims to generate a rational approach to identifying potential targets and effective correctors of the F508del-CFTR folding/ trafficking defects. At the same time, we of course agree that any therapeutic intervention should combine the correction of proteostasis, with the restoration of the activity of the channel; and that the restitution of channel conductance at the PM that we observe does not always match up to the level of rescue obtained on proteostasis alone. This is probably because the chloride channel activity is a complex phenomenon that is determined by several factors in addition to proteostasis correction, and includes regulation by signaling pathways (for e.g. PKA pathway) and the presence (or absence) of regulators such as NHERF and cytoskeletal components. Further studies that reveal the regulatory network controlling the activity of the channel at the PM need to be performed to complement our study on proteostasis and will help in the rational design of pharmaceutical approaches.

As suggested by the reviewers, we have now further emphasized in the paper that one needs to be cautious when assessing the clinical relevance of our results. We have discussed this in the Discussion, which reads as follows:

“A remaining obstacle is the need to combine the correction of proteostasis, which is the focus of this manuscript, with the restoration of the activity of the channel. […] Further studies that reveal the regulatory network controlling the activity of the channel at the PM will complement our study on proteostasis and will help in the rational design of pharmaceutical approaches.”

*7) The authors extensively use siRNA to investigate the effect of knocking down specific transcripts on expression of mature CFTR. siRNAs are notorious for off-target effects, and cannot be assumed to active unless this is directly shown. This problem is illustrated in Figure 3, where the two negative control siRNAs already show drastic differences in band B, although both are presumably not affecting any transcript. For any major conclusions in the paper, on-target effect needs to be quantified for the siRNAs used, and off-target effects need to be excluded by either showing dose-dependent effect of a series of siRNAs with increasing knockdown efficiency, or rescue of the wild-type phenotype by expressing a knockdown-resistant version of the transcript.*

We have carried out the experiments requested by the reviewers. In summary, our confidence that the observed proteostatic effects are specific and not due to off-target effect of the siRNAs is based on the following lines of evidence: 1) Quantitation of the on-target effect of the siRNAs by quantitative RT-PCR (see Figure 2—figure supplement 3), which shows that treatment with siRNAs brings down the transcript levels of the target genes to about 10-30% of the levels present in cells treated with non-targeting siRNAs. 2) We have used at least 3 different siRNAs (individually) for the screenings. Moreover, for selected genes (MLK3, CAMKK2, RNF215, NUP50 and CD2BP2) the findings from the screening studies were reinforced using additional siRNAs (for instance, 5 additional siRNAs for MLK3; see [Supplementary-material SD6-data] for details of the siRNAs used) that showed similar effects on proteostasis of F508del-CFTR (not shown). 3) The coherent effects of the known signaling pathway components on proteostasis that suggests that the observed effects of kinase inhibition are not due to off-target effects. 4) The results presented in Figure 3 where the overexpression of the MLK3 pathway proteins show an opposite effect on proteostasis to that of the siRNAs further confirms the specificity of the effect of the siRNAs used. 5) The control suggested by the reviewers where we have done a dose-dependency of the effect of a set of siRNAs (targeting RNF215, CD2BP2, MLK3 or JNK2) and found a positive correlation between the concentration of siRNAs used and the effect on proteostasis (Figure 2—figure supplement 3). This is now discussed in the Methods section (see “Biochemical screening assay”).

*8) In the Introduction, the authors state that "systematic exploration of the signalling pathways that regulate the initial stages of the proteostasis viz. the folding and degradation of proteins is lacking." This is ignoring a large body of work into the cellular signalling pathways associated with cytosolic proteostasis (heat shock response) and the ER (unfolded protein response, UPR). The therapeutic potential of targeting these pathways is actively being explored by other groups (e.g. see PMID: 23647985 for a review). Please add.*

While we agree with the reviewers that UPR and HSR can be defined signaling pathways, we think that they are of a different kind, and have different cellular functions by comparison with the signaling reactions that are traditionally studied in the signaling research area.

The signaling field has historically focused on reactions, in many cases phosphorylation cascades that are activated by extracellular signals and plasma membrane receptors and regulate most, if not all, of the cellular modules. Most of the known drugs are directed against specific components, very often membrane receptors and kinases that are involved in these signaling pathways. The main function of these pathways is to change the functional configuration of the cell.

The HSR and UPR are instead intracellular reactions that operate to maintain the cellular protein homeostasis, i.e. address imbalances between the load of unfolded proteins and the folding capacity of a cell, mostly by enhancing the transcription of the cellular folding machinery. Therefore, there has been a very logical attempt by several researchers to induce these reactions by pharmacological means with the aim to potentiate the folding machinery and so improve the folding of conformationally defective mutants such as F508del-CFTR.

Our statement on signaling was not meant to diminish or neglect the outstanding work of the above-mentioned researchers on UPR and HSR. It was to point to the fact that there is a major field, rich in druggable targets, which might be very fruitful to explore in the field of conformational diseases. In this context, it might be useful to maintain the distinction between the two types of regulatory mechanisms.

Also of note is that, while the UPR and HSR can affect F508del-CFTR proteostasis, they seem to lack in specificity towards different classes of proteins, unlike the signaling pathways described here. Such specificity allows the regulation of the folding-trafficking machinery operating on a protein class, without affecting other classes, with obvious potential therapeutic advantages.

We have added a note in the Introduction to highlight the distinctions discussed above and to accommodate the reviewers’ suggestions. The text now reads as follows:

“In contrast to the fairly extensive knowledge about the machinery involved in the proteostasis (or protein homeostasis) of F508del-CFTR, the regulatory mechanisms that operate on the F508del-CFTR proteostasis machinery remain relatively less explored. […] Identifying the relevant regulatory components of these systems would not only enhance our understanding of the physiology of proteostasis, but also have significant impact on future therapeutic developments, because components of the signalling cascades, such as membrane receptors and kinases, are generally druggable, and are, in fact, the main targets of most known drugs.”

The Abstract is also modified to include these considerations:

“Cystic fibrosis (CF) is caused by mutations in CF transmembrane conductance regulator (CFTR). The most frequent mutation (F508del-CFTR) results in altered proteostasis, i.e., in the misfolding and intracellular degradation of the protein. The F508del-CFTR proteostasis machinery and its homeostatic regulation are well studied, while the question whether ‘classical’ signalling pathways and phosphorylation cascades might control proteostasis remains barely explored.”

9) Also in the Introduction, the authors state that "Unfortunately, the effects of the available correctors are too weak to be of clinical interest". However, Lumacaftor (VX-809) in combination with ivacaftor is FDA approved for F508delta CF patients, based on efficacy in clinical trials (improved lung function over placebo): http://www.fda.gov/NewsEvents/Newsroom/PressAnnouncements/ucm453565.htm

We apologize for not mentioning this previously. The Lumacaftor study has now been discussed in the Introduction. It reads as follows:

“The mechanism of action (MOA) of the pharmacochaperones has been partially understood (Farinha et al., 2013; Okiyoneda et al., 2013), and these molecules are approaching the level of effectiveness required for clinical use [(Wainwright et al., 2015) and see also http://www.fda.gov/NewsEvents/Newsroom/PressAnnouncements/ucm453565.htm].”

*10) The authors conclude that increase in CFTR levels cannot be due to increased synthesis based on the unchanged mRNA levels. They cannot exclude effects on translational efficiency.*

We agree with the reviewers that we cannot technically exclude an effect on translation. However, when appreciated in light of our observations – that MLK3 pathway regulates the degradation of F508del-CFTR both at the level of ER and PM – we think that the regulation is more at the degradation level of the protein than at the synthesis.

Nevertheless, we have modified the Results section to read as follows:

“It is also to be noted that the downregulation of anti-correction genes did not change the levels of F508del-CFTR mRNA (Figure 2—figure supplement 2) suggesting that the observed effect is not due to increased F508del-CFTR synthesis, although these data cannot by themselves exclude an effect on translation. Further experiments confirm that the effect of the downregulation of these genes is mostly (if not completely) on F508del-CFTR degradation and folding/export (see below).”

“The increase in the levels of band B induced by inhibition of the MLK3 pathway might be due to increased synthesis or due to decreased degradation of F508del-CFTR. Downregulation of MLK3 did not increase the CFTR mRNA levels (Figure 2—figure supplement 2), speaking against the former possibility, though an effect of this pathway on the translational efficiency cannot be excluded.”

*11) Figure 2: triangle for mitoxantrone concentration appears flipped? Actual concentrations should be indicated in the figure or legend.*

No, the triangle for mitoxantrone concentration is not flipped. The apparent reduction in the effect of mitoxantrone at higher concentrations we propose could be due its off-target effects. The range of concentrations used (2.5 to 20µM) is mentioned in the figure legend.

*12) Official gene symbols are not consistently used, and labels for siRNAs in figures do not always correspond to the labels in [Supplementary-material SD5-data], where the siRNA sequences are listed.*

We have now appropriately modified [Supplementary-material SD5-data] to reflect the official gene symbol and common names used in the text. We have not used the official gene symbol in the main text and referred to the genes/proteins using their common names to enhance the readability of the text. The official genes symbols are nevertheless mentioned when gene/protein is discussed for the first time in the manuscript.

*13) The trypsin susceptibility assay may capture some, but not all differences in folding that are relevant for ERAD/QC. In general, the evidence the authors provide to characterize the mechanism and nature of MLK3 regulation should be presented with all the relevant caveats.*

We agree with the reviewer, and in fact, we think that, on balance, the collective evidence indicates that MLK3 has a positive effect on F508del-CFTR folding/export. To reflect this, we have made appropriate changes to the text and also discussed the possibility of an effect of the MLK3 pathway on folding-export.

The Results section of the manuscript now reads as follows:

“The trypsin susceptibility assay to assess the folding status of F508del-CFTR and an assay for protein transport out of the ER using vesicular stomatitis virus G protein (VSVG), a classical probe to study secretory trafficking, ruled out large effects of the MLK3 pathway on F508del-CFTR folding or on the general ER-export machinery (Figure 4—figure supplement 1). Nevertheless, we note that these assays (trypsin susceptibility or VSVG export) are limited in their scope and do not capture the wide spectrum of subtle regulations that can influence the outcome of proteostasis (see below for a discussion of the effect of the MLK3 pathway on folding/export).”

Additionally, please see the following passage of the Discussion:

“Although MLK3 does not measurably affect the folding of F508del-CFTR as measured by the trypsin susceptibility assay, it cannot be excluded, however, that MLK3 (and other CORE genes) might exert subtle direct actions on the folding and/or ER export mechanisms. This is supported by the strong effects of some of the CORE pathways on the band C/band B ratio, and also by the observation that the inhibition of MLK3 markedly stimulates the efficiency of export of a mutant of ATP7B (similar in structure to CFTR) from the ER (Chesi et al., submitted).”

*14) The finding that siRNA knockdown of TAK1 does not equal the effect of oxozeaenol is not by itself proof that oxozeaenol does not act through TAK1, since pharmacological inhibition and siRNA knockdown of kinases can have distinct consequences.*

We agree with the reviewers that pharmacological inhibition and siRNA knockdown of kinases can have different consequences, for instance regarding the catalytic and non-catalytic activities of a kinase or the presence of splice variants or isoforms. Since the siRNAs we used were targeting all the known isoforms, we think the simplest interpretation for the absence of an effect of TAK1 siRNA on F508del-proteostasis is that oxozeaenol does not act through TAK1.

We have now modified the Results section to incorporate this point:

“However, the downregulation of TAK1 itself had no effect on correction (Figure 6—figure supplement 1). Although the pharmacological inhibition and the siRNA knockdown of kinases can sometimes have distinct consequences, these data suggest that oxozeaenol likely acts through MLK3 pathway kinases to affect proteostasis rather than through TAK1. In line with this notion, the corrective effects of oxozeaenol were not additive with MLK3 knockdown (Figure 6—figure supplement 1) and were accompanied by a reduction in phospho c-jun levels (c-jun phosphorylation is diagnostic of JNK activity) (Figure 6—figure supplement 1).”

*15) It would be nice but is not essential to show this efficacy in primary lung tissue from patients. It is well established that the influence of the proteostasis regulators on F508del-CFTR folding and trafficking is not always repeatable when translating the work from easily culturable cells to lung cells derived from patients-the standard now used by those commercial organizations developing CF drugs.*

As discussed in the manuscript, translating the work to therapy is beyond our scope here and we do agree with the reviewers that a study of the effect of these drugs on primary lung tissue from patients will be necessary to transpose this work into therapy. We have discussed these considerations in the Discussion section under the title “Relevance of the CORE signalling networks for the development of clinically useful proteostasis correctors.”

*16) It also seems important to show (or at least discuss) that the 20-fold increases in the C-band has a functional significance in the presence of a channel opening pharmacologic agent. The F508del-CFTR is deficient as a chloride channel even when it is properly folded and trafficked to the plasma membrane of polarized epithelial cells.*

As discussed earlier, the goal of this study is to elucidate the signaling networks that control the proteostasis machinery acting on F508del-CFTR, with the intent to fill a knowledge gap and provide a basis for rational ways to identify major potential targets and effective F508del-CFTR correctors. This study does not aim to generate efficient correctors ready for clinical use; hence, we did not focused on rescuing the activity of the channel itself.

Having said that, our correctors, in addition to rescuing the deficient proteostasis of F508del-CFTR, also rescue the functional activity of the channel, as measured by an increase in the chloride channel conductance at the PM. However, these assays also revealed a quantitative discrepancy between the effect on proteostasis and the effect on chloride channel activity, i.e. a 20-fold increase in Band C levels corresponds only to 4-5 fold increase in chloride channel activity at the PM, even with the use of the potentiator VX-770. Regulation of the chloride channel activity is a complex phenomenon and is determined by several factors including proteostasis pathways, signaling pathways (for e.g. PKA pathway) and the presence (or absence) of regulators such as NHERF and cytoskeletal components. The apparent discrepancy between the effect on proteostasis and the effect on chloride channel activity [which has been observed before (see Hutt et al. 2010) suggests that restoration of proteostasis alone may not be enough to functionally restore chloride channel activity at the PM. Further studies that reveal the regulatory network controlling the activity of the channel at the PM will complement our study on proteostasis and will help in the rational design of pharmaceutical approaches. As suggested by the reviewer, we have now discussed this issue in the Discussion, as stated before.